# CLASS NORMALIZATION FOR (CONTINUAL)? GENERALIZED ZERO-SHOT LEARNING

Ivan Skorokhodov[1,2]
Thuwal, Saudi Arabia
iskorokhodov@gmail.com

Mohamed Elhoseiny[1]
Thuwal, Saudi Arabia
mohamed.elhoseiny@kaust.edu.sa

[1]King Abdullah University of Science and Technology (KAUST), Saudi Arabia
[2]Moscow Institute of Physics and Technology (MIPT), Russia

## ABSTRACT

Normalization techniques have proved to be a crucial ingredient of successful training in a traditional supervised learning regime. However, in the zero-shot learning (ZSL) world, these ideas have received only marginal attention. This work studies normalization in ZSL scenario from both theoretical and practical perspectives. First, we give a theoretical explanation to two popular tricks used in zero-shot learning: normalize+scale and attributes normalization and show that they help training by preserving variance during a forward pass. Next, we demonstrate that they are insufficient to normalize a deep ZSL model and propose *Class Normalization (CN)*: a normalization scheme, which alleviates this issue both provably and in practice. Third, we show that ZSL models typically have more irregular loss surface compared to traditional classifiers and that the proposed method partially remedies this problem. Then, we test our approach on 4 standard ZSL datasets and outperform sophisticated modern SotA with a simple MLP optimized without any bells and whistles and having ≈50 times faster training speed. Finally, we generalize ZSL to a broader problem — continual ZSL, and introduce some principled metrics and rigorous baselines for this new setup. The source code is available at https://github.com/universome/class-norm.

## 1 INTRODUCTION

Zero-shot learning (ZSL) aims to understand new concepts based on their semantic descriptions instead of numerous input-output learning pairs. It is a key element of human intelligence and our best machines still struggle to master it (Ferrari & Zisserman, 2008; Lampert et al., 2009; Xian et al., 2018a). Normalization techniques like batch/layer/group normalization (Ioffe & Szegedy, 2015; Ba et al., 2016; Wu & He, 2018) are now a common and important practice of modern deep learning. But despite their popularity in traditional supervised training, not much is explored in the realm of zero-shot learning, which motivated us to study and investigate normalization in ZSL models.

We start by analyzing two ubiquitous tricks employed by ZSL and representation learning practitioners: *normalize+scale (NS)* and *attributes normalization (AN)* (Bell et al., 2016; Zhang et al., 2019; Guo et al., 2020; Chaudhry et al., 2019). Their dramatic influence on performance can be observed from Table 1. When these two tricks are employed, a vanilla MLP model, described in Sec 3.1, can outperform some recent sophisticated ZSL methods.

**Normalize+scale (NS)** changes logits computation from usual dot-product to *scaled* cosine similarity:

$$\hat{y}_c = \boldsymbol{z}^\top \boldsymbol{p}_c \Longrightarrow \hat{y}_c = \left( \gamma \cdot \frac{\boldsymbol{z}}{\|\boldsymbol{z}\|_2} \right)^\top \left( \gamma \cdot \frac{\boldsymbol{p}_c}{\|\boldsymbol{p}_c\|} \right) \tag{1}$$

where $\boldsymbol{z}$ is an image feature, $\boldsymbol{p}_c$ is $c$-th class prototype and $\gamma$ is a hyperparameter, usually picked from $[5, 10]$ interval (Li et al., 2019; Zhang et al., 2019). Scaling by $\gamma$ is equivalent to setting a high temperature of $\gamma^2$ in softmax. In Sec. 3.2, we theoretically justify the need for this trick and explain why the value of $\gamma$ must be so high.

Table 1: **Effectiveness of Normalize+Scale, Attributes Normalization and Class Normalization**. When NS and AN are integrated into a basic ZSL model, its performance is boosted up to a level of some sophisticated SotA methods and additionally using CN allows to outperform them. ±NS and ±AN denote if normalize+scale or attributes normalization are being used. Bold/normal blue font denote best/second-best results. Extended results are in Table 2, 5 and 8.

| | SUN | | | CUB | | | AwA1 | | | AwA2 | | | Avg training time |
|---|---|---|---|---|---|---|---|---|---|---|---|---|---|
| | U | S | H | U | S | H | U | S | H | U | S | H | |
| DCN Liu et al. (2018) | 25.5 | 37.0 | 30.2 | 28.4 | 60.7 | 38.7 | - | - | - | 25.5 | 84.2 | 39.1 | 50 minutes |
| SGAL Yu & Lee (2019) | 42.9 | 31.2 | 36.1 | 47.1 | 44.7 | 45.9 | 52.7 | 75.7 | 62.2 | 55.1 | 81.2 | 65.6 | 50 minutes |
| LsrGAN Vyas et al. (2020) | 44.8 | 37.7 | 40.9 | 48.1 | 59.1 | **53.0** | - | - | - | 54.6 | 74.6 | 63.0 | 1.25 hours |
| Vanilla MLP -NS -AN | 4.7 | 27.2 | 8.0 | 5.9 | 26.0 | 9.7 | 43.1 | 81.3 | 56.3 | 37.7 | 84.3 | 52.1 | |
| Vanilla MLP -NS +AN | 9.6 | 34.0 | 14.9 | 8.8 | 4.6 | 6.0 | 28.6 | 84.4 | 42.7 | 23.3 | 87.4 | 36.8 | 30 seconds |
| Vanilla MLP +NS -AN | 34.7 | 38.5 | 36.5 | 46.9 | 42.8 | 44.9 | 57.0 | 69.9 | 62.8 | 49.7 | 76.4 | 60.2 | |
| Vanilla MLP +NS+AN | 31.4 | 40.4 | 35.3 | 45.2 | 50.7 | 47.8 | 58.1 | 70.3 | 63.6 | 58.2 | 73.0 | 64.8 | |
| Vanilla MLP +NS +AN +**CN** | 44.7 | 41.6 | **43.1** | 49.9 | 50.7 | 50.3 | 63.1 | 73.4 | **67.8** | 60.2 | 77.1 | **67.6** | 30 seconds |

**Attributes Normalization (AN)** technique simply divides class attributes by their $L_2$ norms:

$$\boldsymbol{a}_c \longmapsto \boldsymbol{a}_c/\|\boldsymbol{a}_c\|_2 \tag{2}$$

While this may look inconsiderable, it is surprising to see it being preferred in practice (Li et al., 2019; Narayan et al., 2020; Chaudhry et al., 2019) instead of the traditional zero-mean and unit-variance data standardization (Glorot & Bengio, 2010). In Sec 3, we show that it helps in normalizing signal's variance in and ablate its importance in Table 1 and Appx D.

These two tricks work well and normalize the variance to a unit value when the underlying ZSL model is linear (see Figure 1), but they fail when we use a multi-layer architecture. To remedy this issue, we introduce *Class Normalization (CN)*: a novel normalization scheme, which is based on a different initialization and a class-wise standardization transform. Modern ZSL methods either utilize sophisticated architectural design like training generative models (Narayan et al., 2020; Felix et al., 2018) or use heavy optimization schemes like episode-based training (Yu et al., 2020; Li et al., 2019). In contrast, we show that simply adding Class Normalization on top of a vanilla MLP is enough to set new state-of-the-art results on several standard ZSL datasets (see Table 2). Moreover, since it is optimized with plain gradient descent without any bells and whistles, training time for us takes *50-100 times less and runs in about 1 minute.* We also demonstrate that many ZSL models tend to have more irregular loss surface compared to traditional supervised learning classifiers and apply the results of Santurkar et al. (2018) to show that our CN partially remedies the issue. We discuss and empirically validate this in Sec 3.5 and Appx F.

Apart from the theoretical exposition and a new normalization scheme, we also propose a broader ZSL setup: continual zero-shot learning (CZSL). Continual learning (CL) is an ability to acquire new knowledge without forgetting (e.g. (Kirkpatrick et al., 2017)), which is scarcely investigated in ZSL. We develop the ideas of lifelong learning with class attributes, originally proposed by Chaudhry et al. (2019) and extended by Wei et al. (2020a), propose several principled metrics for it and test several classical CL methods in this new setup.

## 2  RELATED WORK

**Zero-shot learning**. Zero-shot learning (ZSL) aims at understand example of unseen classes from their language or semantic descriptions. Earlier ZSL methods directly predict attribute confidence from images to facilitate zero-shot recognition (e.g., Lampert et al. (2009); Farhadi et al. (2009); Lampert et al. (2013b)). Recent ZSL methods for image classification can be categorized into two groups: generative-based and embedding-based. The main goal for generative-based approaches is to build a conditional generative model (e.g., GANs Goodfellow et al. (2014) and VAEs (Kingma & Welling, 2014)) to synthesize visual generations conditioned on class descriptors (e.g., Xian et al. (2018b); Zhu et al. (2018); Elhoseiny & Elfeki (2019); Guo et al. (2017); Guo et al. (2017); Kumar Verma et al. (2018)). At test time, the trained generator is expected to produce synthetic/fake data of unseen classes given its semantic descriptor. The fake data is then used to train a traditional classifier or to perform a simple kNN-classification on the test images. Embedding-based approaches learn a mapping that projects semantic attributes and images into a common space where the distance

between a class projection and the corresponding images is minimized (e.g, Romera-Paredes & Torr (2015); Frome et al. (2013); Lei Ba et al. (2015); Akata et al. (2016a); Zhang et al. (2017); Akata et al. (2015; 2016b)). One question that arises is what space to choose to project the attributes or images to. Previous works projected images to the semantic space (Elhoseiny et al., 2013; Frome et al., 2013; Lampert et al., 2013a) or some common space (Zhang & Saligrama, 2015; Akata et al., 2015), but our approach follows the idea of Zhang et al. (2016); Li et al. (2019) that shows that projecting attributes to the image space reduces the bias towards seen data.

**Normalize+scale and attributes normalization**. It was observed both in ZSL (e.g., Li et al. (2019); Zhang et al. (2019); Bell et al. (2016)) and representation learning (e.g., Sohn (2016); Guo et al. (2020); Ye et al. (2020)) fields that normalize+scale (i.e. (1)) and attributes normalization (i.e. (2)) tend to significantly improve the performance of a learning system. In the literature, these two techniques lack rigorous motivation and are usually introduced as practical heuristics that aid training (Changpinyo et al., 2017; Zhang et al., 2019; 2021). One of the earliest works that employ attributes normalization was done by (Norouzi et al., 2013), and in (Changpinyo et al., 2016a) authors also ablate its importance. The main consumers of normalize+scale trick had been similarity learning algorithms, which employ it to refine the distance metric between the representations (Bellet et al., 2013; Guo et al., 2020; Shi et al., 2020). Luo et al. (2018) proposed to use cosine similarity in the final output projection matrix as a normalization procedure, but didn't incorporate any analysis on how it affects the variance. They also didn't use the scaling which our experiments in Table 5 show to be crucial. Gidaris & Komodakis (2018) demonstrated a greatly superior performance of an NS-enriched model compared to a dot-product based one in their setup where the classifying matrix is constructed dynamically. Li et al. (2019) motivated their usage of NS by variance reduction, but didn't elaborate on this in their subsequent analysis. Chen et al. (2020) related the use of the normalized temperature-scaled cross entropy loss (NT-Xent) to different weighting of negative examples in contrastive learning framework. Overall, to the best of our knowledge, there is no precise understanding of the influence of these two tricks on the optimization process and benefits they provide.

**Initialization schemes**. In the seminal work, Xavier's init Glorot & Bengio (2010), the authors showed how to preserve the variance during a forward pass. He et al. (2015) applied a similar analysis but taking ReLU nonlinearities into account. There is also a growing interest in two-step Jia et al. (2014), data-dependent Krähenbühl et al. (2015), and orthogonal Hu et al. (2020) initialization schemes. However, the importance of a good initialization for multi-modal embedding functions like attribute embedding is less studied and not well understood. We propose a proper initialization scheme based on a different initialization variance and a dynamic standardization layer. Our variance analyzis is similar in nature to Chang et al. (2020) since attribute embedder may be seen as a hypernetwork (Ha et al., 2016) that outputs a linear classifier. But the exact embedding transformation is different from a hypernetwork since it has matrix-wise input and in our derivations we have to use more loose assumptions about attributes distribution (see Sec 3 and Appx H).

**Normalization techniques**. A closely related branch of research is the development of normalization layers for deep neural networks (Ioffe & Szegedy, 2015) since they also influence a signal's variance. BatchNorm, being the most popular one, normalizes the location and scale of activations. It is applied in a batch-wise fashion and that's why its performance is highly dependent on batch size (Singh & Krishnan, 2020). That's why several normalization techniques have been proposed to eliminate the batch-size dependecy (Wu & He, 2018; Ba et al., 2016; Singh & Krishnan, 2020). The proposed class normalization is very similar to a standardization procedure which underlies BatchNorm, but it is applied class-wise in the attribute embedder. This also makes it independent from the batch size.

**Continual zero-shot learning**. We introduce continual zero-shot learning: a new benchmark for ZSL agents that is inspired by continual learning literature (e.g., Kirkpatrick et al. (2017)). It is a development of the scenario proposed in Chaudhry et al. (2019), but authors there focused on ZSL performance only a single task ahead, while in our case we consider the performance on all seen (previous tasks) and all unseen data (future tasks). This also contrasts our work to the very recent work by Wei et al. (2020b), where a sequence of seen class splits of existing ZSL benchmsks is trained and the zero-shot performance is reported for every task individually at test time. In contrast, for our setup, the label space is not restricted and covers the spectrum of all previous tasks (seen tasks so far), and future tasks (unseen tasks so far). Due to this difference, we need to introduce a set of new metrics and benchmarks to measure this continual generalized ZSL skill over time. From the lifelong learning perspective, the idea to consider all the processed data to evaluate the model is not

new and was previously explored by Elhoseiny et al. (2018); van de Ven & Tolias (2019). It lies in contrast with the common practice of providing task identity at test time, which limits the prediction space for a model, making the problem easier (Kirkpatrick et al., 2017; Aljundi et al., 2017). In Isele et al. (2016); Lopez-Paz & Ranzato (2017) authors motivate the use of task descriptors for zero-shot knowledge transfer, but in our work we consider class descriptors instead. We defined CZSL as a continual version of generalized-ZSL which allows us to naturally extend all the existing ZSL metrics Xian et al. (2018a); Chao et al. (2016) to our new continual setup.

## 3 NORMALIZATION IN ZERO-SHOT LEARNING

The goal of a good normalization scheme is to preserve a signal inside a model from severe fluctuations and to keep it in the regions that are appropriate for subsequent transformations. For example, for *ReLU* activations, we aim that its input activations to be zero-centered and not scaled too much: otherwise, we risk to find ourselves in all-zero or all-linear activation regimes, disrupting the model performance. For *logits*, we aim them to have a close-to-unit variance since too small variance leads to poor gradients of the subsequent cross-entropy loss and too large variance is an indicator of poor scaling of the preceding weight matrix. For *linear* layers, we aim their inputs to be zero-centered: in the opposite case, they would produce too biased outputs, which is undesirable.

In traditional supervised learning, we have different normalization and initialization techniques to control the signal flow. In zero-shot learning (ZSL), however, the set of tools is extremely limited. In this section, we justify the popularity of Normalize+Scale (NS) and Attributes Normalization (AN) techniques by demonstrating that they just retain a signal variance. We demonstrate that they are not enough to normalize a deep ZSL model and propose class normalization to regulate a signal inside a deep ZSL model. We empirically evaluate our study in Sec. 5 and appendices A, B, D and F.

### 3.1 NOTATION

A ZSL setup considers access to datasets of seen and unseen images with the corresponding labels $D^s = \{\boldsymbol{x}_i^s, y_i^s\}_{i=1}^{N_s}$ and $D^u = \{\boldsymbol{x}_i^u, y_i^u\}_{i=1}^{N_u}$ respectively. Each class $c$ is described by its class attribute vector $\boldsymbol{a}_c \in \mathbb{R}^{d_a}$. All attribute vectors are partitioned into non-overlapping seen and unseen sets as well: $A^s = \{\boldsymbol{a}_i\}_{i=1}^{K_s}$ and $A^u = \{\boldsymbol{a}_i\}_{i=1}^{K_u}$. Here $N_s, N_u, K_s, K_u$ are number of seen images, unseen images, seen classes, and unseen classes respectively. In modern ZSL, all images are usually transformed via some standard feature extractor $E : \boldsymbol{x} \mapsto \boldsymbol{z} \in \mathbb{R}^{d_z}$ (Xian et al., 2018a). Then, a typical ZSL method trains *attribute embedder* $P_{\boldsymbol{\theta}} : \boldsymbol{a}_c \to \boldsymbol{p}_c \in \mathbb{R}^{d_z}$ which projects class attributes $\boldsymbol{a}_c^s$ onto feature space $\mathbb{R}^{d_z}$ in such a way that it lies closer to exemplar features $\boldsymbol{z}^s$ of its class $c$.

This is done by solving a classification task, where logits are computed using formula (1). In such a way at test time we are able to classify unseen images by projecting unseen attribute vectors $\boldsymbol{a}_c^u$ into the feature space and computing similarity with the provided features $\boldsymbol{z}^u$. Attribute embedder $P_{\boldsymbol{\theta}}$ is usually a very simple neural network (Li et al., 2019); in many cases even linear (Romera-Paredes & Torr, 2015; Elhoseiny et al., 2013), so it is the training procedure and different regularization schemes that carry the load. We will denote the final projection matrix and the body of $P_{\boldsymbol{\theta}}$ as $V$ and $H_{\boldsymbol{\varphi}}$ respectively, i.e. $P_{\boldsymbol{\theta}}(\boldsymbol{a}_c) = V H_{\boldsymbol{\varphi}}(\boldsymbol{a}_c)$. During training, it receives matrix of class attributes $A = [\boldsymbol{a}_1, ..., \boldsymbol{a}_{K_s}]$ of size $K_s \times d_a$ and outputs matrix $W = P_{\boldsymbol{\theta}}(A)$ of size $K_s \times d_z$. Then $W$ is used to compute class logits with a batch of image feature vectors $\boldsymbol{z}_1, ..., \boldsymbol{z}_{N_s}$.

### 3.2 UNDERSTANDING NORMALIZE + SCALE TRICK

One of the most popular *tricks* in ZSL and deep learning is using the scaled cosine similarity instead of a simple dot product in logits computation (Li et al., 2019; Zhang et al., 2019; Ye et al., 2020):

$$\hat{y}_c = \boldsymbol{z}^\top \boldsymbol{p}_c \implies \hat{y}_c = \gamma^2 \frac{\boldsymbol{z}^\top \boldsymbol{p}_c}{\|\boldsymbol{z}\|\|\boldsymbol{p}_c\|} \tag{3}$$

where hyperparameter $\gamma$ is usually picked from $[5, 10]$ interval. Both using the cosine similarity and scaling it afterwards by a large value is critical to obtain good performance; see Appendix D. To our knowledge, it has not been clear why exactly it has such big influence and why the value of $\gamma$ must be so large. The following statement provides an answer to these questions.

**Statement 1 (informal)**. *Normalize+scale trick forces the variance for $\hat{y}_c$ to be approximately*:

$$\text{Var}\left[\hat{y}_c\right] \approx \gamma^4 \frac{d_z}{(d_z - 2)^2}, \tag{4}$$

*where $d_z$ is the dimensionality of the feature space*. See Appendix A for the assumptions, derivation and the empirical study. Formula (4) demonstrates two things:

1. When we use cosine similarity, the variance of $\hat{y}_c$ becomes independent from the variance of $W = P_{\boldsymbol{\theta}}(A)$, leading to better stability.

2. If one uses Eq. (3) without scaling (i.e. $\gamma = 1$), then the $\text{Var}\left[\hat{y}_c\right]$ will be extremely low (especially for large $d_z$) and our model will always output uniform distribution and the training would stale. That's why we need very large values for $\gamma$.

Usually, the optimal value of $\gamma$ is found via a hyperparameter search (Li et al., 2019), but our formula suggests another strategy: one can obtain any desired variance $\nu = \text{Var}\left[\hat{y}_c\right]$ by setting $\gamma$ to:

$$\gamma = \left(\frac{\nu \cdot (d_z - 2)^2}{d_z}\right)^{\frac{1}{4}} \tag{5}$$

For example, for $\text{Var}\left[\hat{y}_c\right] = \nu = 1$ and $d_z = 2048$ we obtain $\gamma \approx 6.78$, which falls right in the middle of $[5, 10]$ — a usual search region for $\gamma$ used by ZSL and representation learning practitioners Li et al. (2019); Zhang et al. (2019); Guo et al. (2020). The above consideration not only gives a theoretical understanding of the trick, which we believe is important on its own right, but also allows to speed up the search by either picking the predicted "optimal" value for $\gamma$ or by searching in its vicinity.

### 3.3 Understanding attributes normalization trick

We showed in the previous subsection that "normalize+scale" trick makes the variances of $\hat{y}_c$ independent from variance of weights, features and attributes. This may create an impression that it does not matter how we initialize the weights — normalization would undo any fluctuations. However it is not true, because it is still important how the signal flows under the hood, i.e. for an unnormalized and unscaled logit value $\tilde{y}_c = \boldsymbol{z}^\top \boldsymbol{p}_c$. Another common trick in ZSL is the normalization of attribute vectors to a unit norm $\boldsymbol{a}_c \longmapsto \frac{\boldsymbol{a}_c}{\|\boldsymbol{a}_c\|_2}$. We provide some theoretical underpinnings of its importance.

Let's first consider a linear case for $P_{\boldsymbol{\theta}}$, i.e. $H_{\boldsymbol{\varphi}}$ is an identity, thus $\tilde{y}_c = \boldsymbol{z}^\top \boldsymbol{p}_c = \boldsymbol{z}^\top V \boldsymbol{a}_c$. Then, the way we initialize $V$ is crucial since $\text{Var}\left[\tilde{y}\right]_c$ depends on it. To derive an initialization scheme people use 3 strong assumptions for the inputs Glorot & Bengio (2010); He et al. (2015); Chang et al. (2020): 1) they are zero-centered 2) independent from each other; and 3) have the covariance matrix of the form $\sigma^2 I$. But in ZSL setting, we have two sources of inputs: image features $\boldsymbol{z}$ and class attributes $\boldsymbol{a}_c$. And these assumptions are safe to assume only for $\boldsymbol{z}$ but not for $\boldsymbol{a}_c$, because they do not hold for the standard datasets (see Appendix H). To account for this, we derive the variance $\text{Var}\left[\hat{y}_c\right]$ without relying on these assumptions for $\boldsymbol{a}_c$ (see Appendix B):

$$\text{Var}\left[\tilde{y}_c\right] = d_z \cdot \text{Var}\left[z_i\right] \cdot \text{Var}\left[V_{ij}\right] \cdot \mathbb{E}_{\boldsymbol{a}}\left[\|\boldsymbol{a}\|_2^2\right] \tag{6}$$

From equation (6) one can see that after giving up invalid assumptions for $\boldsymbol{a}_c$, pre-logits variance $\text{Var}\left[\tilde{y}_c\right]$ now became dependent on $\|\boldsymbol{a}_c\|_2$, which is not captured by traditional Glorot & Bengio (2010) and He et al. (2015) initialization schemes and thus leads to poor variance control. Attributes normalization trick rectifies this limitation, which is summarized in the following statement.

**Statement 2 (informal)**. *Attributes normalization trick leads to the same pre-logits variance as we have with Xavier fan-out initialization*. (see Appendix B for the formal statement and the derivation).

Xavier fan-out initialization selects such a scale for a linear layer that the variance of backward pass representations is preserved across the model (in the absence of non-linearities). The fact that attributes normalization results in the scaling of $P_{\boldsymbol{\theta}}$ equivalent to Xavier fan-out scaling and not some other one is a coincidence and shows what underlying meaning this procedure has.

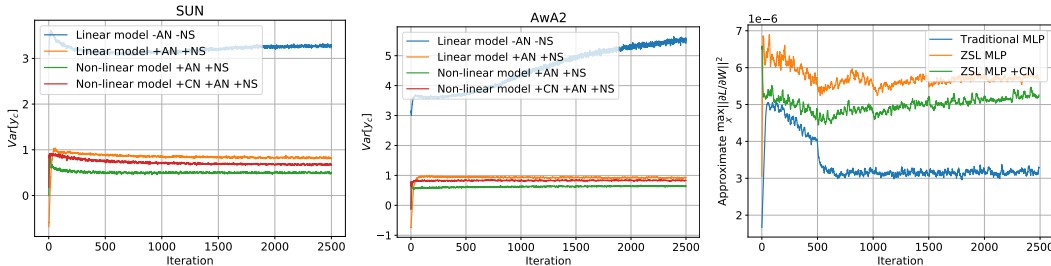

Figure 1: Logits variances (two left plots) and the approximate loss landscape smoothness (right plot) measured during training for different models. From variances plots, one can observe the following picture: a linear model without NS (normalize+scale) and AN (attributes normalization) has diverging variance, but adding NS+AN fixes this pushing the variance to 1. For a non-linear model, using NS and AN on their own is not enough and the variance deteriorates, but class normalization (CN) rectifies it back to 1; see additional analysis in Appx E. On the right plot, the maximum gradient magnitude over 10 batches at a given iteration for different classification models is presented. As stated in 3.5, ZSL attribute embedders have more irregular loss surface than traditional models: large gradient norms indicate abrupt changes in the landscape. Class normalization makes it more smooth; see more analysis in Appx F.

## 3.4 CLASS NORMALIZATION

What happens when $P_{\boldsymbol{\theta}}$ is not linear? Let $\boldsymbol{h}_c = H_{\boldsymbol{\varphi}}(\boldsymbol{a}_c)$ be the output of $H_{\boldsymbol{\varphi}}$. The analysis of this case is equivalent to the previous one but with plugging in $\boldsymbol{h}_c$ everywhere instead of $\boldsymbol{a}_c$. This lead to:

$$\text{Var}\left[\tilde{y}_c\right] = d_z \cdot \text{Var}\left[z_i\right] \cdot \text{Var}\left[V_{ij}\right] \cdot \mathop{\mathbb{E}}_{\boldsymbol{h}}\left[\|\boldsymbol{h}\|_2^2\right] \tag{7}$$

As a result, to obtain $\text{Var}\left[\tilde{y}_c\right] = \text{Var}\left[z_i\right]$ property, we need to initialize $\text{Var}\left[V_{ij}\right]$ the following way:

$$\text{Var}\left[V_{ij}\right] = \left(d_z \cdot \mathop{\mathbb{E}}_{\boldsymbol{h}_c}\left[\|\boldsymbol{h}_c\|_2^2\right]\right)^{-1} \tag{8}$$

This makes the initialization dependent on the magnitude of $\|\boldsymbol{h}_c\|$ instead of $\|\boldsymbol{a}_c\|$, so normalizing attributes to a unit norm would not be sufficient to preserve the variance. To initialize the weights of $V$ using this formula, a two-step data-dependent initialization is required: first initializing $H_{\boldsymbol{\varphi}}$, then computing average $\|\boldsymbol{h}_c\|_2^2$, and then initializing $V$. However, this is not reliable since $\|\boldsymbol{h}_c\|_2^2$ changes on each iteration, so we propose a more elegant solution to standardize $\boldsymbol{h}_c$

$$S(\boldsymbol{h}_c) = (\boldsymbol{h}_c - \hat{\boldsymbol{\mu}})/\hat{\boldsymbol{\sigma}} \tag{9}$$

As one can note, this is similar to BatchNorm standardization without the subsequent affine transform, but we apply it *class-wise* on top of attribute embeddings $\boldsymbol{h}_c$. We plug it in right before $V$, i.e. $P_{\boldsymbol{\theta}}(\boldsymbol{a}_c) = V S(H_{\boldsymbol{\varphi}}(\boldsymbol{a}_c))$. This does not add any parameters and has imperceptible computational overhead. At test time, we use statistics accumulated during training similar to batch norm. Standardization (9) makes inputs to $V$ have constant norm, which now makes it trivial to pick a proper value to initialize $V$:

$$\mathop{\mathbb{E}}_{\boldsymbol{h}_c}\left[\|S(\boldsymbol{h}_c)\|_2^2\right] = d_h \implies \text{Var}\left[V_{ij}\right] = \frac{1}{d_z d_h}. \tag{10}$$

We coin the simultaneous use of (9) and (10) *class normalization* and highlight its influence in the following statement. See Fig. 3 for the model diagram, Fig. 1 for empirical study of its impact, and Appendix C for the assumptions, proof and additional details.

**Statement 3 (informal)**. *Standardization procedure (9) together with the proper variance formula (10), preserves the variance between $\boldsymbol{z}$ and $\tilde{y}$ for a mutli-layer attribute embedder $P_{\boldsymbol{\theta}}$.*

## 3.5 IMPROVED SMOOTHNESS

We also analyze the loss surface smoothness for $P_{\boldsymbol{\theta}}$. There are many ways to measure this notion (Hochreiter & Schmidhuber, 1997; Keskar et al., 2016; Dinh et al., 2017; Skorokhodov & Burtsev,

2019), but following Santurkar et al. (2018), we define it in a "per-layer" fashion via Lipschitzness:

$$g^\ell = \max_{\|X^{\ell-1}\|_2 \leq \lambda} \|\nabla_{W^\ell} \mathcal{L}\|_2^2, \tag{11}$$

where $\ell$ is the layer index and $X^{\ell-1}$ is its input data matrix. This definition is intuitive: larger gradient magnitudes indicate that the loss surface is prone to abrupt changes. We demonstrate two things:

1. For each example in a batch, parameters of a ZSL attribute embedder receive $K$ more updates than a typical non-ZSL classifier, where $K$ is the number of classes. This suggests a hypothesis that it has larger overall gradient magnitude, hence a more irregular loss surface.

2. Our standardization procedure (9) makes the surface more smooth. We demonstrate it by simply applying Theorem 4.4 from (Santurkar et al., 2018).

Due to the space constraints, we defer the exposition on this to Appendix F.

## 4 CONTINUAL ZERO-SHOT LEARNING

### 4.1 PROBLEM FORMULATION

In continual learning (CL), a model is being trained on a sequence of tasks that arrive one by one. Each task is defined by a dataset $D^t = \{\boldsymbol{x}_i^t, y_i^t\}_{i=1}^{N_t}$ of size $N_t$. The goal of the model is to learn all the tasks sequentially in such a way that at each task $t$ it has good performance both on the current task and all the previously observed ones. In this section we develop the ideas of Chaudhry et al. (2019) and formulate a Continual Zero-Shot Learning (CZSL) problem. Like in CL, CZSL also assumes a sequence of tasks, but now each task is a *generalized* zero-shot learning problem. This means that apart from $D^t$ we also receive a set of corresponding class descriptions $A^t$ for each task $t$. In this way, traditional zero-shot learning can be seen as a special case of CZSL with just two tasks. In Chaudhry et al. (2019), authors evaluate their zero-shot models on each task individually, without considering the classification space across tasks; looking only one step ahead, which gives a limited picture of the model's quality. Instead, we borrow ideas from Generalized ZSL (Chao et al., 2016; Xian et al., 2018a), and propose to measure the performance on all the seen and all the unseen data for each task. More formally, for timestep $t$ we have the datasets:

$$D^{\leq t} = \bigcup_{r=1}^{t} D^r \qquad D^{>t} = \bigcup_{r=t+1}^{T} D^r \qquad A^{\leq t} = \bigcup_{r=1}^{t} A^r \qquad A^{>t} = \bigcup_{r=t+1}^{T} A^r \tag{12}$$

which are the datasets of all seen data (learned tasks), all unseen data (future tasks), seen class attributes, and unseen class attributes respectively. For our proposed CZSL, the model at timestep $t$ has access to only data $D^t$ and attributes $A^t$, but its goal is to have good performance on all seen data $D^{\leq t}$ and all unseen data $D^{>t}$ with the corresponding attributes sets $A^{\leq t}$ and $A^{>t}$. For $T = 2$, this would be equivalent to traditional generalized zero-shot learning. But for $T > 2$, it is a novel and a much more challenging problem.

### 4.2 PROPOSED EVALUATION METRICS

Our metrics for CZSL use GZSL metrics under the hood and are based on *generalized accuracy* (GA) (Chao et al., 2016; Xian et al., 2018a). "Traditional" seen (unseen) accuracy computation discards unseen (seen) classes from the prediction space, thus making the problem easier, since the model has fewer classes to be distracted with. For generalized accuracy, we always consider the joint space of both seen and unseen and this is how GZSL-S and GZSL-U are constructed. We use this notion to construct *mean seen* (mS), *mean unseen* (mU) and *mean harmonic* (mH) accuracies. We do that just by measuring *GZSL-S/GZSL-U/GZSL-H* at each timestep, considering all the past data as seen and all the future data as unseen. Another set of CZSL metrics are *mean joint accuracy* (mJA) which measures the performance across all the classes and *mean area under seen/unseen curve* (mAUC) which is an adaptation of AUSUC measure by Xian et al. (2018a). A more rigorous formulation of these metrics is presented in Appendix G.2. Apart from them, we also employ a popular forgetting measure (Lopez-Paz & Ranzato, 2017).

## 5 EXPERIMENTS

### 5.1 ZSL EXPERIMENTS

**Experiment details**. We use 4 standard datasets: SUN (Patterson et al., 2014), CUB (Welinder et al., 2010), AwA1 and AwA2 and seen/unseen splits from Xian et al. (2018a). They have 645/72, 150/50, 40/10 and 40/10 seen/unseen classes respectively with $d_a$ being equal to 102, 312, 85 and 85 respectively. Following standard practice, we use ResNet101 image features (with $d_z = 2048$) from Xian et al. (2018a). Our attribute embedder $P_\theta$ is a vanilla 3-layer MLP augmented with standardization procedure 9 and corrected output matrix initialization 10. For all the datasets, we train the model with Adam optimizer for 50 epochs and evaluate it at the end of training. We also employ NS and AN techniques with $\gamma = 5$ for NS. Additional hyperparameters are reported in Appx D. To perform cross-validation, we first allocate 10% of seen classes for a *validation unseen* data (for AwA1 and AwA2 we allocated 15% since there are only 40 seen classes). Then we allocate 10% out of the remaining 85% of the data for *validation seen* data. This means that in total we allocate $\approx 30\%$ of all the seen data to perform validation. It is known (Xian et al., 2018a; Min et al., 2020), that GZSL-H score can be improved slightly by reducing the weight of seen class logits during accuracy computation since this would partially relieve the bias towards seen classes. We also employ this trick by multiplying seen class logits by value $s$ during evaluation and find its optimal value using cross-validation together with the other hyperparameters. On Figure 4 in Appendix D.4, we provide validation/test accuracy curves of how it influences the performance.

**Evaluation and discussion**. We evaluate the model on the corresponding test sets using 3 metrics as proposed by Xian et al. (2018a): seen generalized unseen accuracy (GZSL-U), generalized seen accuracy (GZSL-S) and GZSL-S/GZSL-U harmonic mean (GZSL-H), which is considered to the main metric for ZSL. Table 2 shows that our model has the state-of-the-art in 3 out of 4 datasets.

**Training speed results**. We conducted a survey and rerun several recent SotA methods from their official implementations to check their training speed, which details we report in Appx D. Table 2 shows the average training time for each of the methods. Since our model is just a vanilla MLP and does not use any sophisticated training scheme, it trains from 30 to 500 times faster compared to other methods, while outperforming them in the final performance.

### 5.2 CZSL EXPERIMENTS

**Datasets**. We test our approach in CZSL scenario on two datasets: CUB Welinder et al. (2010) and SUN Patterson et al. (2014). CUB contains 200 classes and is randomly split into 10 tasks with 20

Table 2: **Generalized Zero-Shot Learning results**. S, U denote generalized seen/unseen accuracy and H is their harmonic mean. Bold/normal blue font denotes the best/second-best result.

| | SUN | | | CUB | | | AwA1 | | | AwA2 | | | Avg training time |
|---|---|---|---|---|---|---|---|---|---|---|---|---|---|
| | U | S | H | U | S | H | U | S | H | U | S | H | |
| DCN (Liu et al., 2018) | 25.5 | 37.0 | 30.2 | 28.4 | 60.7 | 38.7 | - | - | - | 25.5 | 84.2 | 39.1 | 50 min |
| RN (Sung et al., 2018) | - | - | - | 38.1 | 61.4 | 47.0 | 31.4 | 91.3 | 46.7 | 30.9 | 93.4 | 45.3 | 35 min |
| f-CLSWGAN (Xian et al., 2018b) | 42.6 | 36.6 | 39.4 | 57.7 | 43.7 | 49.7 | 57.9 | 61.4 | 59.6 | - | - | - | - |
| CIZSL (Elhoseiny & Elfeki, 2019) | - | - | 27.8 | - | - | - | - | - | - | - | - | 24.6 | 2 hours |
| CVC-ZSL (Li et al., 2019) | 36.3 | 42.8 | 39.3 | 47.4 | 47.6 | 47.5 | 62.7 | 77.0 | 69.1 | 56.4 | 81.4 | 66.7 | 3 hours |
| SGMA (Zhu et al., 2019) | - | - | - | 36.7 | 71.3 | 48.5 | - | - | - | 37.6 | 87.1 | 52.5 | - |
| SGAL (Yu & Lee, 2019) | 42.9 | 31.2 | 36.1 | 47.1 | 44.7 | 45.9 | 52.7 | 75.7 | 62.2 | 55.1 | 81.2 | 65.6 | 50 min |
| DASCN (Ni et al., 2019) | 42.4 | 38.5 | 40.3 | 45.9 | 59.0 | 51.6 | 59.3 | 68.0 | 63.4 | - | - | - | - |
| F-VAEGAN-D2 (Xian et al., 2019) | 45.1 | 38.0 | 41.3 | 48.4 | 60.1 | 53.6 | - | - | - | 57.6 | 70.6 | 63.5 | - |
| TF-VAEGAN (Narayan et al., 2020) | 45.6 | 40.7 | 43.0 | 52.8 | 64.7 | **58.1** | - | - | - | 59.8 | 75.1 | 66.6 | 1.75 hours |
| EPGN (Yu et al., 2020) | - | - | - | 52.0 | 61.1 | 56.2 | 62.1 | 83.4 | **71.2** | 52.6 | 83.5 | 64.6 | - |
| DVBE (Min et al., 2020) | 45.0 | 37.2 | 40.7 | 53.2 | 60.2 | 56.5 | - | - | - | 63.6 | 70.8 | 67.0 | - |
| LsrGAN (Vyas et al., 2020) | 44.8 | 37.7 | 40.9 | 48.1 | 59.1 | 53.0 | - | - | - | 54.6 | 74.6 | 63.0 | 1.25 hours |
| ZSML (Verma et al., 2020) | - | - | - | 60.0 | 52.1 | 55.7 | 57.4 | 71.1 | 63.5 | 58.9 | 74.6 | 65.8 | - |
| 3-layer MLP | 31.4 | 40.4 | 35.3 | 45.2 | 48.4 | 46.7 | 57.0 | 69.9 | 62.8 | 54.5 | 72.2 | 62.1 | |
| 3-layer MLP + Eq. (9) | 41.5 | 41.3 | 41.4 | 49.4 | 48.6 | 49.0 | 60.1 | 73.0 | 65.9 | 60.3 | 75.6 | 67.1 | 30 seconds |
| 3-layer MLP + Eq. (10) | 24.1 | 37.9 | 29.5 | 45.3 | 44.5 | 44.9 | 58.4 | 70.7 | 64.0 | 52.1 | 72.0 | 60.5 | |
| 3-layer MLP + **CN** (i.e. (9) + (10)) | 44.7 | 41.6 | **43.1** | 49.9 | 50.7 | 50.3 | 63.1 | 73.4 | 67.8 | 60.2 | 77.1 | **67.6** | |

Table 3: Continual Zero-Shot Learning results with and without CN. Best scores are in bold blue.

| | CUB | | | | SUN | | | |
|---|---|---|---|---|---|---|---|---|
| | mAUC ↑ | mH ↑ | mJA ↑ | Forgetting ↓ | mAUC↑ | mH↑ | mJA↑ | Forgetting ↓ |
| EWC-online (Schwarz et al., 2018) | 11.6 | 18.0 | 25.4 | 0.08 | 2.7 | 9.6 | 11.4 | **0.02** |
| EWC-online + ClassNorm | **14.1**$^{+22\%}$ | 23.3$^{+29\%}$ | **28.6**$^{+13\%}$ | **0.04**$^{-50\%}$ | **4.8**$^{+78\%}$ | **14.3**$^{+49\%}$ | **15.8**$^{+39\%}$ | 0.03$^{+50\%}$ |
| MAS-online (Aljundi et al., 2017) | 11.4 | 17.7 | 25.1 | 0.08 | 2.5 | 9.4 | 11.0 | **0.02** |
| MAS-online + ClassNorm | 14.0$^{+23\%}$ | **23.8**$^{+34\%}$ | 28.5$^{+14\%}$ | 0.05$^{-37\%}$ | **4.8**$^{+92\%}$ | 14.2$^{+51\%}$ | **15.8**$^{+44\%}$ | 0.03$^{+50\%}$ |
| A-GEM (Chaudhry et al., 2019) | 10.4 | 17.3 | 23.6 | 0.16 | 2.4 | 9.6 | 10.8 | 0.05 |
| A-GEM + ClassNorm | 13.8$^{+33\%}$ | **23.8**$^{+38\%}$ | 28.2$^{+19\%}$ | 0.06$^{-62\%}$ | 4.6$^{+92\%}$ | 14.2$^{+48\%}$ | 15.4$^{+43\%}$ | 0.04$^{-20\%}$ |
| Sequential | 9.7 | 17.2 | 22.6 | 0.17 | 2.3 | 9.3 | 10.4 | 0.05 |
| Sequential + ClassNorm | 13.5$^{+39\%}$ | 23.0$^{+34\%}$ | 27.9$^{+23\%}$ | 0.05$^{-71\%}$ | 4.6$^{+99\%}$ | 14.0$^{+51\%}$ | 15.3$^{+47\%}$ | 0.03$^{-40\%}$ |
| Multi-task | 23.4 | 24.3 | 39.6 | 0.00 | 4.2 | 12.5 | 14.9 | 0.00 |
| Multi-task + ClassNorm | 26.5$^{+13\%}$ | 30.0$^{+23\%}$ | 42.6$^{+8\%}$ | 0.01 | 6.2$^{+48\%}$ | 14.8$^{+18\%}$ | 18.5$^{+24\%}$ | 0.01 |

classes per task. SUN contains 717 classes which is randomly split into 15 tasks, the first 3 tasks have 47 classes and the rest of them have 48 classes each (717 classes are difficult to separate evenly). We use official train/test splits for training and testing the model.

**Model and optimization**. We follow the proposed cross-validation procedure from Chaudhry et al. (2019). Namely, for each run we allocate the first 3 tasks for hyperparameter search, validating on the test data. After that we reinitialize the model from scratch, discard the first 3 tasks and train it on the rest of the data. This reduces the effective number of tasks by 3, but provides a more fair way to perform cross-validation Chaudhry et al. (2019). We use an ImageNet-pretrained ResNet-18 model as an image encoder $E(\boldsymbol{x})$ which is optimized jointly with $P_{\boldsymbol{\theta}}$. For CZSL experiments, $P_{\boldsymbol{\theta}}$ is a 2-layer MLP and we test the proposed CN procedure. All the details can be found in Appendix G.

We test our approach on 3 continual learning methods: EWC Kirkpatrick et al. (2017), MAS Aljundi et al. (2017) and A-GEM Chaudhry et al. (2019) and 2 benchmarks: Multi-Task model and Sequential model. EWC and MAS fight forgetting by regularizing the weight update for a new task in such a way that the important parameters are preserved. A-GEM maintains a memory bank of previously encountered examples and performs a gradient step in such a manner that the loss does not increase on them. Multi-Task is an "upper bound" baseline: a model which has an access to all the previously encountered data and trains on them jointly. Sequential is a "lower bound" baseline: a model which does not employ any technique at all. We give each model an equal number of update iterations on each task. This makes the comparison of the Multi-Task baseline to other methods more fair: otherwise, since its dataset grows with time, it would make $t$ times more updates inside task $t$ than the other methods.

**Evaluation and discussion**. Results for the proposed metrics mU, mS, mH, mAUC, mJA and forgetting measure from Lopez-Paz & Ranzato (2017) are reported in Table 3 and Appendix G. As one can observe, class normalization boosts the performance of classical regularization-based and replay-based continual learning methods by up to 100% and leads to lesser forgetting. However, we are still far behind traditional supervised classifiers as one can infer from mJA metric. For example, some state-of-the-art approaches on CUB surpass 90% accuracy Ge et al. (2019) which is drastically larger compared to what the considered approaches achieve.

## 6 CONCLUSION

We investigated and developed normalization techniques for zero-shot learning. We provided theoretical groundings for two popular tricks: normalize+scale and attributes normalization and showed both provably and in practice that they aid training by controlling a signal's variance during a forward pass. Next, we demonstrated that they are not enough to constrain a signal from fluctuations for a deep ZSL model. That motivated us to develop class normalization: a new normalization scheme that fixes the problem and allows to obtain SotA performance on 4 standard ZSL datasets in terms of quantitative performance and training speed. Next, we showed that ZSL attribute embedders tend to have more irregular loss landscape than traditional classifiers and that class normalization partially remedies this issue. Finally, we generalized ZSL to a broader setting of continual zero-shot learning and proposed a set of principled metrics and baselines for it. We believe that our work will spur the development of stronger zero-shot systems and motivate their deployment in real-world applications.

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

# A "NORMALIZE + SCALE" TRICK

As being discussed in Section 3.2, "normalize+scale" trick changes the logits computation from a usual dot product to the scaled cosine similarity:

$$\hat{y}_c = \langle \boldsymbol{z}, \boldsymbol{p}_c \rangle \implies \hat{y}_c = \left\langle \gamma \frac{\boldsymbol{z}}{\|\boldsymbol{z}\|}, \gamma \frac{\boldsymbol{p}_c}{\|\boldsymbol{p}_c\|} \right\rangle, \tag{13}$$

where $\hat{y}_c$ is the logit value for class $c$; $\boldsymbol{z}$ is an image feature vector; $\boldsymbol{p}_c$ is the attribute embedding for class $c$:

$$\boldsymbol{p}_c = P_{\boldsymbol{\theta}}(\boldsymbol{a}_c) = V H_{\boldsymbol{\varphi}(\boldsymbol{a}_c)} \tag{14}$$

and $\gamma$ is the scaling hyperparameter. Let's denote a penultimate hidden representation of $P_{\boldsymbol{\theta}}$ as $\boldsymbol{h}_c = H_{\boldsymbol{\varphi}}(\boldsymbol{a}_c)$. We note that in case of linear $P_{\boldsymbol{\theta}}$, we have $\boldsymbol{h}_c = \boldsymbol{a}_c$. Let's also denote the dimensionalities of $\boldsymbol{z}$ and $\boldsymbol{h}_c$ by $d_z$ and $d_h$.

## A.1 ASSUMPTIONS

To derive the approximate variance formula for $\hat{y}_c$ we will use the following assumptions and approximate identities:

(i) All weights in matrix $V$:
- are independent from each other and from $z_k$ and $h_{c,i}$ (for all $k, i$);
- $\mathbb{E}[V_{ij}] = 0$ for all $i, j$;
- $\mathrm{Var}[V_{ij}] = s_v$ for all $i, j$.

(ii) There exists $\epsilon > 0$ s.t. $(2 + \epsilon)$-th central moment exists for each of $h_{c,1}, ..., h_{c,d_h}$. We require this technical condition to be able apply the central limit theorem for variables with non-equal variances.

(iii) All $h_{c,i}, h_{c,j}$ are independent from each other for $i \neq j$. This is the least realistic assumption from the list, because in case of linear $P_{\boldsymbol{\theta}}$ it would be equivalent to independence of coordinates in attribute vector $\boldsymbol{a}_c$. We are not going to use it in other statements. As we show in Appendix A.3 it works well in practice.

(iv) All $p_{c,i}, p_{c,j}$ are independent between each other. This is also a nasty assumption, but more safe to assume in practice (for example, it is easy to demonstrate that $\mathrm{Cov}[p_{c,i}, p_{c,j}] = 0$ for $i \neq j$). We are going to use it only in normalize+scale approximate variance formula derivation.

(v) $\boldsymbol{z} \sim \mathcal{N}(\boldsymbol{0}, s^z I)$. This property is safe to assume since $\boldsymbol{z}$ is usually a hidden representation of a deep neural network and each coordinate is computed as a vector-vector product between independent vectors which results in the normal distribution (see the proof below for $\boldsymbol{p}_c \rightsquigarrow \mathcal{N}(\boldsymbol{0}, s^p I)$).

(vi) For $\boldsymbol{\xi} \in \{\boldsymbol{z}, \boldsymbol{p}_c\}$ we will use the approximations:

$$\mathbb{E}\left[\xi_i \cdot \frac{1}{\|\boldsymbol{\xi}\|_2}\right] \approx \mathbb{E}[\xi_i] \cdot \mathbb{E}\left[\frac{1}{\|\boldsymbol{\xi}\|_2}\right] \quad \text{and} \quad \mathbb{E}\left[\xi_i \xi_j \cdot \frac{1}{\|\boldsymbol{\xi}\|_2^2}\right] \approx \mathbb{E}[\xi_i \xi_j] \cdot \mathbb{E}\left[\frac{1}{\|\boldsymbol{\xi}\|_2^2}\right] \tag{15}$$

This approximation is safe to use if the dimensionality of $\boldsymbol{\xi}$ is large enough (for neural networks it is definitely the case) because the contribution of each individual $\xi_i$ in the norm $\|\boldsymbol{\xi}\|_2$ becomes negligible.

Assumptions (i-v) are typical for such kind of analysis and can also be found in (Glorot & Bengio, 2010; He et al., 2015; Chang et al., 2020). Assumption (vi), as noted, holds only for large-dimensional inputs, but this is exactly our case and we validate that using leads to a decent approximation on Figure 2.

## A.2 FORMAL STATEMENT AND THE PROOF

**Statement 1** (Normalize+scale trick). *If conditions (i)-(vi) hold, then:*

$$\mathrm{Var}[\hat{y}_c] = \mathrm{Var}\left[\left\langle \gamma \frac{\boldsymbol{z}}{\|\boldsymbol{z}\|}, \gamma \frac{\boldsymbol{p}_c}{\|\boldsymbol{p}_c\|} \right\rangle\right] \approx \frac{\gamma^4 d_z}{(d_z - 2)^2} \tag{16}$$

*Proof.* First of all, we need to show that $p_{c,i} \rightsquigarrow \mathcal{N}(0, s^p)$ for some constant $s^p$. Since

$$p_{c,i} = \sum_{j=1}^{d_h} V_{i,j} h_{c,j} \tag{17}$$

from assumption (i) we can easily compute its mean:

$$\mathbb{E}\left[p_{c,i}\right] = \mathbb{E}\left[\sum_{j=1}^{d_h} V_{i,j} h_{c,j}\right] \tag{18}$$

$$= \sum_{j=1}^{d_h} \mathbb{E}\left[V_{i,j} h_{c,j}\right] \tag{19}$$

$$= \sum_{j=1}^{d_h} \mathbb{E}\left[V_{i,j}\right] \cdot \mathbb{E}\left[h_{c,j}\right] \tag{20}$$

$$= \sum_{j=1}^{d_h} 0 \cdot \mathbb{E}\left[h_{c,j}\right] \tag{21}$$

$$= 0. \tag{22}$$

and the variance:

$$\text{Var}\left[p_{c,i}\right] = \mathbb{E}\left[p_{c,i}^2\right] - \left(\mathbb{E}\left[p_{c,i}\right]\right)^2 \tag{23}$$

$$= \mathbb{E}\left[p_{c,i}^2\right] \tag{24}$$

$$= \mathbb{E}\left[\left(\sum_{j=1}^{d_h} V_{i,j} h_{c,j}\right)^2\right] \tag{25}$$

$$= \mathbb{E}\left[\sum_{j,k=1}^{d_h} V_{i,j} V_{i,k} h_{c,j} h_{c,k}\right] \tag{26}$$

Using $\mathbb{E}\left[V_{i,j} V_{i,k}\right] = 0$ for $k \neq j$, we have:

$$= \mathbb{E}\left[\sum_{j}^{d_h} V_{i,j}^2 h_{c,j}^2\right] \tag{27}$$

$$= \sum_{j}^{d_h} \mathbb{E}\left[V_{i,j}^2\right] \mathbb{E}\left[h_{c,j}^2\right] \tag{28}$$

Since $s_v = \text{Var}\left[V_{i,j}\right] = \mathbb{E}\left[V_{i,j}^2\right] - \mathbb{E}\left[V_{i,j}\right]^2 = \mathbb{E}\left[V_{i,j}^2\right]$, we have:

$$= \sum_{j}^{d_h} s_v \mathbb{E}\left[h_{c,j}^2\right] \tag{29}$$

$$= s_c \mathbb{E}\left[\sum_{j}^{d_h} h_{c,j}^2\right] \tag{30}$$

$$= s_v \mathbb{E}\left[\|\boldsymbol{h}_c\|_2^2\right] \tag{31}$$

$$= s^p \tag{32}$$

$$\tag{33}$$

Now, from the assumptions (ii) and (iii) we can apply Lyapunov's Central Limit theorem to $p_{c,i}$, which gives us:

$$\frac{1}{\sqrt{s^p}} p_{c,i} \rightsquigarrow \mathcal{N}(0,1) \tag{34}$$

For finite $d_h$, this allows us say that:

$$p_{c,i} \sim \mathcal{N}(0, s^p) \tag{35}$$

Now note that from (vi) we have:

$$\mathbb{E}\left[\hat{y}_c\right] = \mathbb{E}\left[\left\langle \gamma\frac{\boldsymbol{z}}{\|\boldsymbol{z}\|}, \gamma\frac{\boldsymbol{p}_c}{\|\boldsymbol{p}_c\|} \right\rangle\right] \tag{36}$$

$$= \gamma^2 \left\langle \mathbb{E}\left[\frac{\boldsymbol{z}}{\|\boldsymbol{z}\|}\right], \mathbb{E}\left[\frac{\boldsymbol{p}_c}{\|\boldsymbol{p}_c\|}\right] \right\rangle \tag{37}$$

$$\approx \gamma^2 \mathbb{E}\left[\frac{1}{\|\boldsymbol{z}\|}\right] \cdot \mathbb{E}\left[\frac{1}{\|\boldsymbol{p}_c\|}\right] \cdot \left\langle \mathbb{E}\left[\boldsymbol{z}\right], \mathbb{E}\left[\boldsymbol{p}_c\right]\right\rangle \tag{38}$$

$$= \gamma^2 \mathbb{E}\left[\frac{1}{\|\boldsymbol{z}\|}\right] \cdot \mathbb{E}\left[\frac{1}{\|\boldsymbol{p}_c\|}\right] \cdot \left\langle \boldsymbol{0}, \boldsymbol{0}\right\rangle \tag{39}$$

$$= 0 \tag{40}$$

Since $\boldsymbol{\xi} \sim \mathcal{N}(\boldsymbol{0}, s^\xi)$ for $\boldsymbol{\xi} \in \{\boldsymbol{z}, \boldsymbol{p}_c\}$, $\frac{d_\xi}{\|\boldsymbol{\xi}\|_2^2}$ follows scaled inverse chi-squared distribution with inverse variance $\tau = 1/s^\xi$, which has a known expression for expectation:

$$\mathbb{E}\left[\frac{d_\xi}{\|\boldsymbol{\xi}\|_2^2}\right] = \frac{\tau d_\xi}{d_\xi - 2} = \frac{d_\xi}{s^\xi(d_\xi - 2)} \tag{41}$$

Now we are left with using approximation (vi) and plugging in the above expression into the variance formula:

$$\text{Var}\left[\hat{y}_c\right] = \text{Var}\left[\left\langle \gamma\frac{\boldsymbol{z}}{\|\boldsymbol{z}\|}, \gamma\frac{\boldsymbol{p}_c}{\|\boldsymbol{p}_c\|} \right\rangle\right] \tag{42}$$

$$= \mathbb{E}\left[\left\langle \gamma\frac{\boldsymbol{z}}{\|\boldsymbol{z}\|}, \gamma\frac{\boldsymbol{p}_c}{\|\boldsymbol{p}_c\|} \right\rangle^2\right] - \mathbb{E}\left[\left\langle \gamma\frac{\boldsymbol{z}}{\|\boldsymbol{z}\|}, \gamma\frac{\boldsymbol{p}_c}{\|\boldsymbol{p}_c\|} \right\rangle\right]^2 \tag{43}$$

$$\approx \mathbb{E}\left[\left\langle \gamma\frac{\boldsymbol{z}}{\|\boldsymbol{z}\|}, \gamma\frac{\boldsymbol{p}_c}{\|\boldsymbol{p}_c\|} \right\rangle^2\right] - 0 \tag{44}$$

$$= \gamma^4 \mathbb{E}\left[\frac{(\boldsymbol{z}^\top \boldsymbol{p}_c)^2}{\|\boldsymbol{z}\|^2 \|\boldsymbol{p}_c\|^2}\right] \tag{45}$$

$$\approx \gamma^4 \mathbb{E}\left[(\boldsymbol{z}^\top \boldsymbol{p})^2\right] \cdot \mathbb{E}\left[\frac{1}{d_z} \cdot \frac{d_z}{\|\boldsymbol{z}\|^2}\right] \cdot \mathbb{E}\left[\frac{1}{d_z} \cdot \frac{d_z}{\|\boldsymbol{p}_c\|^2}\right] \tag{46}$$

$$= \frac{\gamma^4}{d_z^2} \cdot \mathbb{E}_{\boldsymbol{p}_c}\left[\boldsymbol{p}_c^\top \mathbb{E}_{\boldsymbol{z}}\left[\boldsymbol{z}\boldsymbol{z}^\top\right] \boldsymbol{p}_c\right] \cdot \frac{d_z}{s^z(d_z - 2)} \cdot \frac{d_z}{s^p(d_z - 2)} \tag{47}$$

$$= \gamma^4 \mathbb{E}_{\boldsymbol{p}_c}\left[\boldsymbol{p}_c^\top s^z I_{d_z} \boldsymbol{p}_c\right] \cdot \frac{1}{s^z s^p (d_z - 2)^2} \tag{48}$$

$$= \gamma^4 \mathbb{E}\left[\sum_{i=1}^{d_z} f_{c,i}^2\right] \cdot \frac{1}{s^p(d_z - 2)^2} \tag{49}$$

$$= \gamma^4 d_z \cdot s^p \cdot \frac{1}{s^p(d_z - 2)^2} \tag{50}$$

$$= \frac{\gamma^4 d_z}{(d_z - 2)^2} \tag{51}$$

$$\tag{52}$$

$$\square$$

## A.3 EMPIRICAL VALIDATION

In this subsection, we validate the derived approximation empirically. For empirical validation of the variance analyzis, see Appendix E. For this, we perform two experiments:

- *Synthetic data.* An experiment on a synthetic data. We sample $\boldsymbol{x} \sim \mathcal{N}(\boldsymbol{0}, I_d), \boldsymbol{y} \sim \mathcal{N}(\boldsymbol{0}, I_d)$ for different dimensionalities $d = 32, 64, 128, ..., 8192$ and compute the cosine similarity:

$$z = \left\langle \frac{\gamma\boldsymbol{x}}{\|\boldsymbol{x}\|}, \frac{\gamma\boldsymbol{y}}{\|\boldsymbol{y}\|} \right\rangle \tag{53}$$

  After that, we compute $\text{Var}\left[z\right]$ and average it out across different samples. The result is presented on figure 2a.

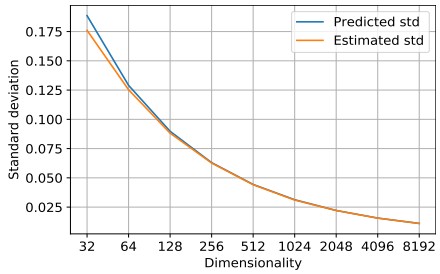 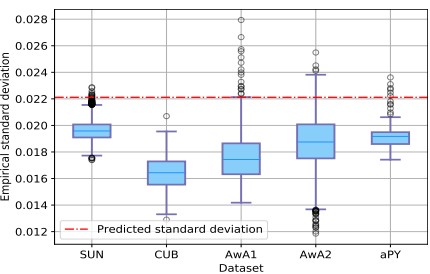

(a) Empirical validation of variance formula (4) on synthetic data

(b) Empirical validation of variance formula (4) on real-world data and for a real-world model

Figure 2: Empirical validation of the derived approximation for variance (4)

- *Real data*. We take ImageNet-pretrained ResNet101 features and real class attributes (unnormalized) for SUN, CUB, AwA1, AwA2 and aPY datasets. Then, we initialize a random 2-layer MLP with 512 hidden units, and generate real logits (without scaling). Then we compute mean empirical variance and the corresponding standard deviation over different batches of size 4096. The resulted boxplots are presented on figure 2b.

In both experiments, we computed the logits with $\gamma = 1$. As one can see, even despite our demanding assumptions, our predicted variance formula is accurate for both synthetic and real-world data.

## B  ATTRIBUTES NORMALIZATION

We will use the same notation as in Appendix A. Attributes normalization trick normalizes attributes to the unit $L_2$-norm:

$$\boldsymbol{a}_c \longmapsto \frac{\boldsymbol{a}_c}{\|\boldsymbol{a}_c\|_2} \tag{54}$$

We will show that it helps to preserve the variance for pre-logit computation when attribute embedder $P_{\boldsymbol{\theta}}$ is linear:

$$\tilde{y}_c = \boldsymbol{z}^\top F(\boldsymbol{a}_c) = \boldsymbol{z}^\top V \boldsymbol{a}_c \tag{55}$$

For a non-linear attribute embedder it is not true, that's why we need the proposed initialization scheme.

### B.1  ASSUMPTIONS

We will need the following assumptions:

    (i) Feature vector $\boldsymbol{z}$ has the properties:
- $\mathbb{E}\left[\boldsymbol{z}\right] = 0$;
- $\text{Var}\left[z_i\right] = s^z$ for all $i = 1, ..., d_z$.
- All $z_i$ are independent from each other and from all $p_{c,j}$.

    (ii) Weight matrix $V$ is initialized with Xavier fan-out mode, i.e. $\text{Var}\left[V_{ij}\right] = 1/d_z$ and are independent from each other.

Note here, that we do not have any assumptions on $\boldsymbol{a}_c$. This is the core difference from Chang et al. (2020) and is an essential condition for ZSL (see Appendix H).

### B.2  FORMAL STATEMENT AND THE PROOF

**Statement 2** (Attributes normalization for a linear embedder). *If assumptions (i)-(ii) are satisfied and $\|\boldsymbol{a}_c\|_2 = 1$, then:*

$$\text{Var}\left[\tilde{y}_c\right] = \text{Var}\left[z_i\right] = s^z \tag{56}$$

*Proof.* Now, note that:

$$\mathbb{E}\left[\tilde{y}_c\right] = \mathbb{E}\left[\boldsymbol{z}^\top V \boldsymbol{a}_c\right] = \underset{V, \boldsymbol{a}_c}{\mathbb{E}}\left[\underset{\boldsymbol{z}}{\mathbb{E}}\left[\boldsymbol{z}^\top\right] V \boldsymbol{a}_c\right] = \mathbb{E}\left[\boldsymbol{0}^\top V \boldsymbol{a}_c\right] = 0 \tag{57}$$

Then the variance for $\tilde{y}_c$ has the following form:

$$\text{Var}\left[\tilde{y}_c\right] = \mathbb{E}\left[\tilde{y}_c^2\right] - \mathbb{E}\left[\tilde{y}_c\right]^2 \tag{58}$$

$$= \mathbb{E}\left[\hat{y}_c^2\right] \tag{59}$$

$$= \mathbb{E}\left[(\boldsymbol{z}^\top V \boldsymbol{a}_c)^2\right] \tag{60}$$

$$= \mathbb{E}\left[\boldsymbol{a}_c^\top V^\top \boldsymbol{z} \boldsymbol{z}^\top V \boldsymbol{a}_c\right] \tag{61}$$

$$= \mathbb{E}_{\boldsymbol{a}_c}\left[\mathbb{E}_V\left[\mathbb{E}_{\boldsymbol{z}}\left[\boldsymbol{a}_c^\top V^\top \boldsymbol{z} \boldsymbol{z}^\top V \boldsymbol{a}_c\right]\right]\right] \tag{62}$$

$$= \mathbb{E}_{\boldsymbol{a}_c}\left[\boldsymbol{a}_c^\top \mathbb{E}_V\left[V^\top \mathbb{E}_{\boldsymbol{z}}\left[\boldsymbol{z} \boldsymbol{z}^\top\right] V\right] \boldsymbol{a}_c\right] \tag{63}$$

$$= s^z \mathbb{E}_{\boldsymbol{a}_c}\left[\boldsymbol{a}_c^\top \mathbb{E}_V\left[V^\top V\right] \boldsymbol{a}_c\right] \tag{64}$$

$$= s^z s^v d_z \mathbb{E}_{\boldsymbol{a}_c}\left[\boldsymbol{a}_c^\top \boldsymbol{a}_c\right] \tag{65}$$

since $s^v = 1/d_z$, then:

$$= s^z \mathbb{E}\left[\|\boldsymbol{a}_c\|_2^2\right] \tag{66}$$

since attributes are normalized, i.e. $\|\boldsymbol{a}_c\|_2 = 1$, then:

$$= s^z \tag{67}$$

$\square$

## C  NORMALIZATION FOR A DEEP ATTRIBUTE EMBEDDER

### C.1  FORMAL STATEMENT AND THE PROOF

Using the same derivation as in B, one can show that for a deep attribute embedder:

$$P_{\boldsymbol{\theta}}(\boldsymbol{a}_c) = V \circ H_{\boldsymbol{\varphi}}(\boldsymbol{a}_c) \tag{68}$$

normalizing attributes is not enough to preserve the variance of $\text{Var}\left[\tilde{y}_c\right]$, because

$$\text{Var}\left[\tilde{y}_c\right] = s^z \mathbb{E}\left[\|\boldsymbol{h}_c\|_2^2\right] \tag{69}$$

and $\boldsymbol{h}_c = H_{\boldsymbol{\varphi}}(\boldsymbol{a}_c)$ is not normalized to a unit norm.

To fix the issue, we are going to use two mechanisms:

1. A different initialization scheme:

$$\text{Var}\left[V_{ij}\right] = \frac{1}{d_z d_h} \tag{70}$$

2. Using the standardization layer before the final projection matrix:

$$S(\boldsymbol{x}) = (\boldsymbol{x} - \hat{\boldsymbol{\mu}}_x) \oslash \hat{\boldsymbol{\sigma}}_x, \tag{71}$$

$\boldsymbol{\mu}_x, \boldsymbol{\sigma}_x$ are the sample mean and variance and $\oslash$ is the element-wise division.

### C.2  ASSUMPTIONS

We'll need the assumption:

(i) Feature vector $\boldsymbol{z}$ has the properties:
   - $\mathbb{E}\left[\boldsymbol{z}\right] = 0$;
   - $\text{Var}\left[z_i\right] = s^z$ for all $i = 1, ..., d_z$.
   - All $z_i$ are independent from each other and from all $p_{c,j}$.

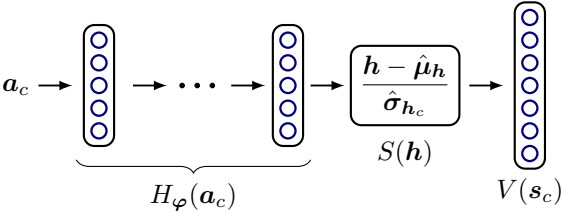

Figure 3: Our architecture: a plain MLP with the standardization procedure (9) inserted before the final projection and output matrix $V$ being initialized using (10).

### C.3 FORMAL STATEMENT AND THE PROOF

**Statement 3.** *If the assumption (i) is satisfied, an attribute embedder has the form $P_{\boldsymbol{\theta}} = V \circ S \circ H_{\boldsymbol{\varphi}}$ and we initialize output matrix $V$ s.t. $\mathrm{Var}\left[V_{ij}\right] = \frac{1}{d_z d_h}$, then the variance for $\tilde{y}_c$ is preserved:*

$$\mathrm{Var}\left[\tilde{y}_c\right] \approx \mathrm{Var}\left[z_i\right] = s^z \tag{72}$$

*Proof.* With some abuse of notation, let $\bar{\boldsymbol{h}}_c = S(\boldsymbol{h}_c)$ (in practice, $S$ receives a batch of $\boldsymbol{h}_c$ instead of a single vector). This leads to:

$$\mathbb{E}\left[\boldsymbol{h}_c\right] = 0 \qquad \text{and} \qquad \mathrm{Var}\left[\boldsymbol{h}_c\right] \approx 1 \tag{73}$$

Using the same reasoning as in Appendix B, one can show that:

$$\mathrm{Var}\left[\tilde{y}_c\right] = d_z \cdot s^z \cdot \mathrm{Var}\left[V_{ij}\right] \cdot \mathbb{E}\left[\|\bar{\boldsymbol{h}}_c\|_2^2\right] = \frac{s^z}{d_h} \cdot \mathbb{E}\left[\|\bar{\boldsymbol{h}}_c\|_2^2\right] \tag{74}$$

So we are left to demonstrate that $\mathbb{E}\left[\|\bar{\boldsymbol{h}}_c\|_2^2\right] = d_h$:

$$\mathbb{E}\left[\|\bar{\boldsymbol{h}}_c\|_2^2\right] = \sum_{i=1}^{d_h} \mathbb{E}\left[\bar{h}_{c,i}^2\right] = \sum_{i=1}^{d_h} \mathrm{Var}\left[\bar{h}_{c,i}\right] \approx d_h \tag{75}$$

### C.4 ADDITIONAL EMPIRICAL STUDIES

$\square$

## D ZSL DETAILS

### D.1 EXPERIMENTS DETAILS

In this section, we cover hyperparameter and training details for our ZSL experiments and also provide the extended training speed comparisons with other methods.

We depict the architecture of our model on figure 3. As being said, $P_{\boldsymbol{\theta}}$ is just a simple multi-layer MLP with standardization procedure (9) and adjusted output layer initialization (10).

Besides, we also found it useful to use entropy regularizer (the same one which is often used in policy gradient methods for exploration) for some datasets:

$$\mathcal{L}_{\mathrm{ent}}(\cdot) = -H(\hat{\boldsymbol{p}}) = \sum_{c=1}^{K} \hat{p}_c \log \hat{p}_c \tag{76}$$

We train the model with Adam optimizer with default $\beta_1$ and $\beta_2$ hyperparams. The list of hyperparameters is presented in Table 4. In those ablation experiments where we do not attributes normalization (2), we apply simple standardization to convert attributes to zero-mean and unit-variance.

### D.2 ADDITIONAL EXPERIMENTS AND ABLATIONS

In this section we present additional experiments and ablation studies for our approach (results are presented in Table 5). We also run validate :

Table 4: Hyperparameters for ZSL experiments

|  | SUN | CUB | AwA1 | AwA2 |
|---|---|---|---|---|
| Batch size | 128 | 512 | 128 | 128 |
| Learning rate | 0.0005 | 0.005 | 0.005 | 0.002 |
| Number of epochs | 50 | 50 | 50 | 50 |
| $\mathcal{L}_{\text{ent}}$ weight | 0.001 | 0.001 | 0.001 | 0.001 |
| Number of hidden layers | 2 | 2 | 2 | 2 |
| Hidden dimension | 2048 | 2048 | 1024 | 512 |
| $\gamma$ | 5 | 5 | 5 | 5 |

Table 5: Additional GZSL ablation studies. From this table one can observe the sensitivity of normalize+scale to $\gamma$ value. We also highlight $\gamma$ importance in Section 3.

|  | SUN | | | CUB | | | AwA1 | | | AwA2 | | |
|---|---|---|---|---|---|---|---|---|---|---|---|---|
|  | U | S | H | U | S | H | U | S | H | U | S | H |
| Linear +NS+AN $\gamma = 1$ | 11.7 | 35.1 | 17.6 | 5.1 | 44.8 | 9.1 | 20.3 | 66.5 | 31.1 | 20.6 | 71.1 | 31.9 |
| Linear +NS+AN $\gamma = 3$ | 11.2 | 37.1 | 17.2 | 10.3 | 53.7 | 17.3 | 13.6 | 72.8 | 23.0 | 21.0 | 50.8 | 29.7 |
| Linear +NS+AN $\gamma = 5$ | 13.8 | 41.0 | 20.6 | 16.8 | 62.0 | 26.4 | 16.8 | 74.2 | 27.4 | 18.9 | 73.2 | 30.0 |
| Linear +NS+AN $\gamma = 10$ | 17.1 | 40.9 | 24.1 | 14.9 | 61.7 | 24.0 | 36.4 | 46.2 | 40.7 | 27.9 | 86.8 | 42.2 |
| Linear +NS+AN $\gamma = 20$ | 13.9 | 35.5 | 20.0 | 13.8 | 52.7 | 21.9 | 46.4 | 43.3 | 44.8 | 47.9 | 59.4 | 53.1 |
| 2-layer MLP +NS+AN $\gamma = 1$ | 34.0 | 36.4 | 35.1 | 38.7 | 37.4 | 38.0 | 49.6 | 61.9 | 55.1 | 49.7 | 65.9 | 56.7 |
| 2-layer MLP +NS+AN $\gamma = 3$ | 32.0 | 37.4 | 34.5 | 42.0 | 43.1 | 42.6 | 53.6 | 68.9 | 60.3 | 53.7 | 72.3 | 61.6 |
| 2-layer MLP +NS+AN $\gamma = 5$ | 34.4 | 39.6 | 36.8 | 46.9 | 45.0 | 45.9 | 57.3 | 73.8 | 64.5 | 55.4 | 77.1 | 64.5 |
| 2-layer MLP +NS+AN $\gamma = 10$ | 31.7 | 37.5 | 34.4 | 47.0 | 43.3 | 45.1 | 54.1 | 65.5 | 59.3 | 56.0 | 72.4 | 63.2 |
| 2-layer MLP +NS+AN $\gamma = 20$ | 56.4 | 11.0 | 18.4 | 44.4 | 35.9 | 39.7 | 51.4 | 69.3 | 59.1 | 46.4 | 73.7 | 56.9 |
| 3-layer MLP +NS+AN $\gamma = 1$ | 18.6 | 37.9 | 25.0 | 23.0 | 42.3 | 29.8 | 50.1 | 60.9 | 55.0 | 48.4 | 64.0 | 55.1 |
| 3-layer MLP +NS+AN $\gamma = 3$ | 23.9 | 37.3 | 29.1 | 35.5 | 48.6 | 41.0 | 57.3 | 67.3 | 61.9 | 57.3 | 70.3 | 63.2 |
| 3-layer MLP +NS+AN $\gamma = 5$ | 31.4 | 40.4 | 35.3 | 45.2 | 50.7 | 47.8 | 58.1 | 70.3 | 63.6 | 58.2 | 73.0 | 64.8 |
| 3-layer MLP +NS+AN $\gamma = 10$ | 29.7 | 37.8 | 33.3 | 40.7 | 40.5 | 40.6 | 55.4 | 63.0 | 58.9 | 53.8 | 69.6 | 60.7 |
| 3-layer MLP +NS+AN $\gamma = 20$ | 15.8 | 39.5 | 22.6 | 22.2 | 54.0 | 31.4 | 53.8 | 63.8 | 58.3 | 49.2 | 69.4 | 57.6 |
| Dynamic Normalization | 31.9 | 39.8 | 35.5 | 22.7 | 56.6 | 32.4 | 58.5 | 68.4 | 63.1 | 55.5 | 70.3 | 62.0 |
| Xavier + (9) | 41.5 | 41.3 | 41.4 | 49.3 | 49.2 | 49.2 | 60.2 | 73.1 | 66.0 | 58.3 | 76.2 | 66.0 |
| Kaiming fan-in + (9) | 42.0 | 41.4 | 41.7 | 51.1 | 49.2 | 50.1 | 59.8 | 74.3 | 66.2 | 55.4 | 75.6 | 63.9 |
| Kaiming fan-out + (9) | 42.8 | 41.2 | 42.0 | 51.0 | 49.0 | 50.0 | 60.3 | 73.2 | 66.1 | 56.8 | 76.9 | 65.4 |

- Dynamic normalization. As one can see from formula (8), to achieve the desired variance it would be enough to initialize $V$ s.t. $\text{Var}\,[V_{ij}] = 1/d_z$ (equivalent to Xavier fan-out) and use a *dynamic normalization*:

$$\text{DN}(\boldsymbol{h}) = \boldsymbol{h}/\mathbb{E}\left[\|\boldsymbol{h}\|_2^2\right] \tag{77}$$

between $V$ and $H_{\boldsymbol{\varphi}}$, i.e. $P_{\boldsymbol{\theta}}(\boldsymbol{a}_c) = V\text{DN}(H_{\boldsymbol{\varphi}}(\boldsymbol{a}_c))$. Expectation $\mathbb{E}\left[\|\boldsymbol{h}\|_2^2\right]$ is computed over a batch on each iteration. A downside of such an approach is that if the dimensionality is large, than a lot of dimensions will get suppressed leading to bad signal propagation. Besides, one has to compute the running statistics to use them at test time which is cumbersome.

- Traditional initializations + standardization procedure (9). These experiments ablate the necessity of using the corrected variance formula (10).

- Performance of NS for different scaling values of $\gamma$ and different number of layers.

## D.3 MEASURING TRAINING SPEED

We conduct a survey and search for open-source implementations of classification ZSL papers that were recently published on top conferences. This is done by 1) checking the papers for code urls; 2) checking their supplementary; 3) searching for implementations on github.com and 4) searching authors by their names on github.com and checking their repositories list. As a result, we found 8 open-source implementations of the recent methods, but one of them got discarded since the corresponding data was not provided. We reran all these

Table 6: Training time for the recent ZSL methods that made their official implementations publicly available. We reran them on the corresponding datasets with the official hyperparameters and training setups. All the comparisons are done on the same machine and hardware: NVidia GeForce RTX 2080 Ti GPU, Intel Xeon Gold 6142 CPU and 64 GB RAM. N/C stands for "no code" meaning that authors didn't release the code for a particular dataset.

|  | SUN | CUB | AwA1 | AwA2 |
|---|---|---|---|---|
| RelationNet Sung et al. (2018) | - | 25 min | 40 min | 40 min |
| DCN Liu et al. (2018) | 40 min | 50 min | - | 55 min |
| CIZSL Elhoseiny & Elfeki (2019) | 3 hours | 2 hours | 3 hours | 3 hours |
| CVC-ZSL Li et al. (2019) | 3 hours | 3 hours | 1.5 hours | 1.5 hours |
| SGAL Yu & Lee (2019) | N/C | N/C | 50 min | N/C |
| LsrGAN Vyas et al. (2020) | 1.1 hours | 1.25 hours | - | 1.5 hours |
| TF-VAEGAN Narayan et al. (2020) | 1.5 hours | 1.75 hours | - | 2 hours |
| Ours | **20 sec** | **20 sec** | **30 sec** | **30 sec** |

methods with the official hyperparameters on the corresponding datasets and report their training time in Table 6 in Appx D.

All runs are made with the official hyperparameters and training setups and on the same hardware: NVidia GeForce RTX 2080 Ti GPU, $\times 16$ Intel Xeon Gold 6142 CPU and 128 GB RAM. The results are depicted on Table 6.

As one can see, the proposed method trains 50-100 faster than the recent SotA. This is due to not using any sophisticated architectures employing generative models (Xian et al., 2018b; Narayan et al., 2020); or optimization schemes like episode-based training (Li et al., 2019; Yu et al., 2020).

### D.4 CHOOSING SCALE $s$ FOR SEEN CLASSES

As mentioned in Section 5, we reweigh seen class logits by multiplying them on scale value $s$. This is similar to a strategy considered by Xian et al. (2018a); Min et al. (2020), but we found that multiplying by a value instead of adding it by summation is more intuitive. We find the optimal scale value $s$ by cross-validation together with all other hyperparameters on the grid $[1.0, 0.95, 0.9, 0.85, 0.8]$. On Figure 4, we depict the curves of how $s$ influences GZSL-U/GZSL-S/GZSL-U for each dataset.

### D.5 INCORPORATING CN FOR OTHER ATTRIBUTE EMBEDDERS

In this section, we employ our proposed class normalization for two other methods: RelationNet[1] (Sung et al., 2018) and CVC-ZSL[2] (Sung et al., 2018). We build upon the officially provided source code bases and use the official hyperparameters for all the setups. For RelationNet, the authors provided the running commands. For CVC-ZSL, we used those hyperparameters for each dataset, that were specified in their paper. That included using different weight decay of $1e-4$, $1e-3$, $1e-3$ and $1e-5$ for AwA1, AwA2, CUB and SUN respectively, as stated in Section 4.2 of the paper (Li et al., 2019). We incorporated our Class Normalization procedure to these attribute embedders and launched them on the corresponding datasets. The results are reported in Table 7. For some reason, we couldn't reproduce the official results for both these methods which we additionally report. As one can see from the presented results, our method gives +2.0 and +1.8 of GZSL-H improvement on average for these two methods respectively which emphasizes once again its favorable influence on the learned representations.

### D.6 ADDITIONAL ABLATION ON AN AND NS TRICKS

In Table 8 we provide additional ablations on attributes normalization and normalize+scale tricks. As one can see, they greatly influence the performance of ZSL attribute embedders.

---

[1] RelationNet: https://github.com/lzrobots/LearningToCompare_ZSL
[2] CVC-ZSL: https://github.com/kailigo/cvcZSL

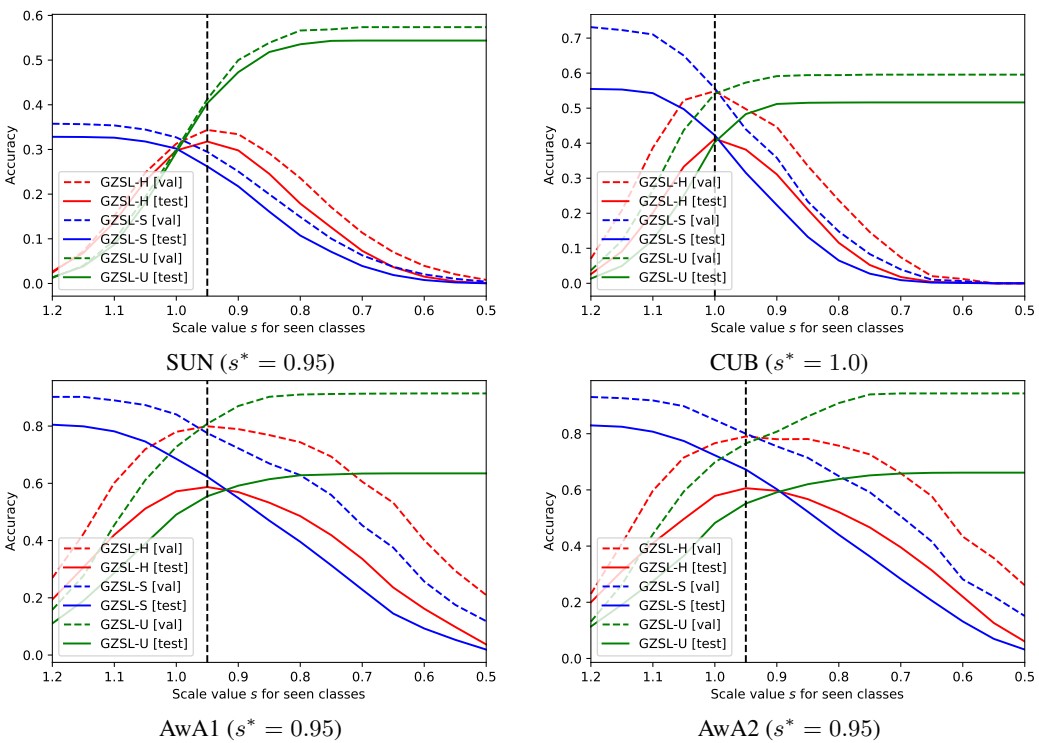

Figure 4: Optimal value of seen logits scale $s$ for different datasets. Multiplying seen logits by some scale $s < 1$ during evaluation leads to sacrificing GZSL-S for an increase in GZSL-U which results in the increased GZSL-H value Xian et al. (2018a); Min et al. (2020). High gap between validation/test accuracy is caused by having different number of classes in these two sets. Lower test GZSL-H than reported in Table 2 is caused by splitting the train set into train/validation sets for the presented run, i.e. using less data for training: we allocated 50, 30, 5 and 5 seen classes for the *validation unseen* ones to construct these plots for SUN, CUB, AwA1 and AwA2 respectively and an equal amount of data was devoted to being used as *validation seen* data, i.e. we "lost" $\approx 25\%$ train data in total. As one can see from these plots, the trick works for those datasets where the gap between GZSL-S and GZSL-U is large and does not give any benefit for CUB where seen and unseen logits are already well-balanced.

Table 7: Incorporating Class Normalization into RelationNet (Sung et al., 2018) and CVC-ZSL (Li et al., 2019) based on the official source code and running hyperparameters. For some reason, our results differ considerably from the reported ones on AwA2 for RelationNet and on SUN for CVC-ZSL. Adding CN provides the improvement in *all* the setups.

| | SUN | | | CUB | | | AwA1 | | | AwA2 | | |
|---|---|---|---|---|---|---|---|---|---|---|---|---|
| | U | S | H | U | S | H | U | S | H | U | S | H |
| RelationNet (official code) | - | - | - | 38.3 | 62.4 | 47.5 | 28.5 | 87.8 | 43.1 | 10.2 | 88.1 | 18.3 |
| RelationNet (official code) + CN | - | - | - | 40.1 | 62.8 | 48.9 | 29.8 | 88.4 | 44.6 | 12.7 | 88.8 | 22.3 |
| CVC-ZSL (official code) | 20.7 | 43.0 | 28.0 | 42.6 | 47.8 | 45.1 | 58.1 | 78.1 | 66.6 | 51.4 | 79.9 | 62.5 |
| CVC-ZSL (official code) + CN | 24.6 | 42.5 | 31.1 | 44.6 | 48.7 | 46.6 | 58.8 | 79.7 | 67.7 | 53.2 | 80.4 | 64.0 |
| RelationNet (reported) | - | - | - | 38.1 | 61.4 | 47.0 | 31.4 | 91.3 | 46.7 | 30.9 | 93.4 | 45.3 |
| CVC-ZSL (reported) | 36.3 | 42.8 | 39.3 | 47.4 | 47.6 | 47.5 | 62.7 | 77.0 | 69.1 | 56.4 | 81.4 | 66.7 |

Table 8: Ablating other methods for AN and NS importance. For CVC-ZSL, we used the officially provided code with the official hyperparameters. When we do not employ AN, we standardize them to zero-mean and unit-variance: otherwise training diverges due to too high attributes magnitudes.

| | SUN | | | CUB | | | AwA1 | | | AwA2 | | |
|---|---|---|---|---|---|---|---|---|---|---|---|---|
| | U | S | H | U | S | H | U | S | H | U | S | H |
| Linear | 41.0 | 33.4 | 36.8 | 26.9 | 58.1 | 36.8 | 40.6 | 76.6 | 53.1 | 38.4 | 81.7 | 52.2 |
| Linear -AN | 13.8 | 41.0 | 20.6 | 16.8 | 62.0 | 26.4 | 16.8 | 74.2 | 27.4 | 18.9 | 73.2 | 30.0 |
| Linear -NS | 38.7 | 3.5 | 6.4 | 33.5 | 6.7 | 11.2 | 44.0 | 42.7 | 43.3 | 44.9 | 48.8 | 46.8 |
| Linear -AN -NS | 17.7 | 2.6 | 4.5 | 3.0 | 0.0 | 0.0 | 13.2 | 0.0 | 0.1 | 23.3 | 0.0 | 0.0 |
| 2-layer MLP | 34.4 | 39.6 | 36.8 | 46.9 | 45.0 | 45.9 | 57.3 | 73.8 | 64.5 | 55.4 | 77.1 | 64.5 |
| 2-layer MLP -AN | 33.8 | 40.1 | 36.7 | 44.9 | 42.9 | 43.9 | 61.9 | 72.5 | 66.8 | 59.1 | 74.1 | 65.8 |
| 2-layer MLP -NS | 51.0 | 11.1 | 18.3 | 40.6 | 22.4 | 28.9 | 40.9 | 68.9 | 51.3 | 40.7 | 69.3 | 51.2 |
| 2-layer MLP -AN -NS | 20.9 | 24.0 | 22.3 | 11.6 | 13.1 | 12.3 | 40.7 | 50.2 | 45.0 | 30.8 | 61.1 | 41.0 |
| CVC-ZSL (official code) | 20.7 | 43.0 | 28.0 | 42.6 | 47.8 | 45.1 | 58.1 | 78.1 | 66.6 | 51.4 | 79.9 | 62.5 |
| CVC-ZSL (official code) -NS | 18.4 | 40.5 | 25.3 | 23.7 | 56.6 | 33.4 | 15.9 | 65.3 | 25.6 | 14.9 | 49.7 | 22.9 |
| CVC-ZSL (official code) -AN | 31.4 | 30.8 | 31.1 | 24.8 | 57.1 | 34.6 | 44.4 | 82.8 | 57.8 | 17.6 | 89.9 | 29.5 |
| CVC-ZSL (official code) -NS -AN | 18.2 | 36.9 | 24.3 | 21.6 | 54.7 | 30.9 | 14.1 | 59.4 | 22.8 | 14.6 | 45.9 | 22.2 |
| CVC-ZSL (reported) | 36.3 | 42.8 | 39.3 | 47.4 | 47.6 | 47.5 | 62.7 | 77.0 | 69.1 | 56.4 | 81.4 | 66.7 |

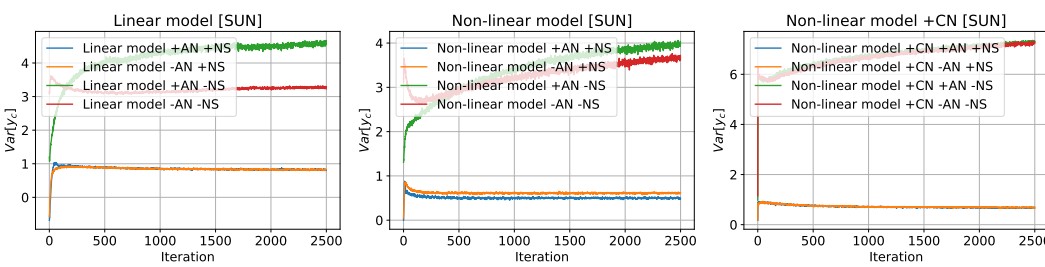

Figure 5: Variances plots for different models for SUN dataset. See Appendix E for the experimental details.

## E  ADDITIONAL VARIANCE ANALYZIS

In this section, we provide the extended variance analyzis for different setups and datasets. The following models are used:

1. A linear ZSL model with/without normalize+scale (NS) and/or attributes normalization (AN).
2. A 3-layer ZSL model with/without NS and/or AN.
3. A 3-layer ZSL model *with class normalization*, with/without NS and/or AN.

These models are trained on 4 standard ZSL datasets: SUN, CUB, AwA1 and AwA2 and their logits variance is calculated on each iteration and reported. The same batch size, learning rate, number of epochs, hidden dimensionalities were used. Results are presented on figures 5, 6, 7 and 8, which illustrates the same trend:

- A traditional linear model without NS and AN has poor variance.
- Adding NS with a proper scaling of 5 and AN improves it and bounds to be close to 1.
- After introducing new layers, NS and AN stop "working" and variance vanishes below unit.
- Incorporating class normalization allows to push it back to 1.

## F  LOSS LANDSCAPE SMOOTHNESS ANALYSIS

### F.1  OVERVIEW

As being said, we demonstrate two things:

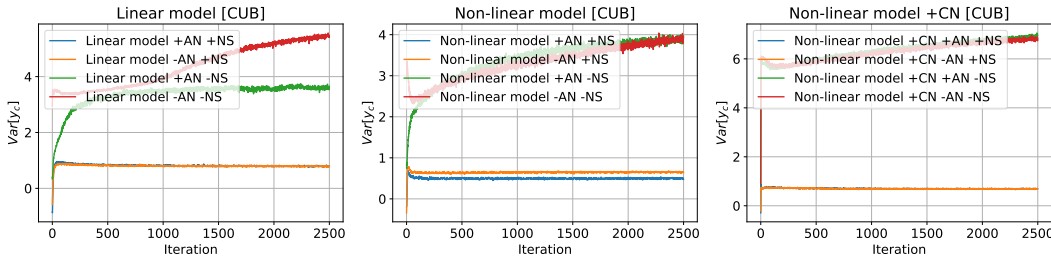

Figure 6: Variances plots for different models for CUB dataset. See Appendix E for the experimental details.

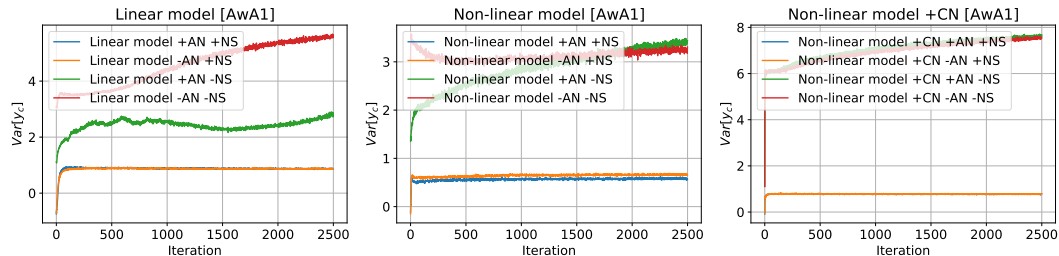

Figure 7: Variances plots for different models for AwA1 dataset. See Appendix E for the experimental details.

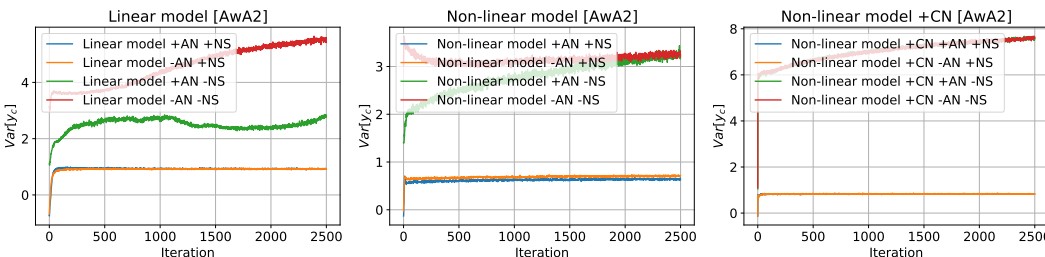

Figure 8: Variances plots for different models for AwA2 dataset. See Appendix E for the experimental details.

1. For each example in a batch, parameters of a ZSL attribute embedder receive $K$ more updates than a typical non-ZSL classifier, where $K$ is the number of classes. This suggests a hypothesis that it has larger overall gradient magnitude, hence a more irregular loss surface.

2. Our standardization procedure (9) makes the surface more smooth. We demonstrate it by simply applying Theorem 4.4 from (Santurkar et al., 2018).

To see the first point, one just needs to compute the derivative with respect to weight $W_{ij}$ for $n$-th data sample for loss surface $\mathcal{L}_n^{\text{SL}}$ of a traditional model and loss surface $\mathcal{L}_n^{\text{ZSL}}$ of a ZSL embedder:

$$\frac{\partial \mathcal{L}_n^{\text{SL}}}{\partial W_{ij}^\ell} = \frac{\partial \mathcal{L}_n^{\text{SL}}}{\partial y_i^{(n)}} x_j^{(n)} \qquad \frac{\partial \mathcal{L}_n^{\text{ZSL}}}{\partial W_{ij}^\ell} = \sum_{c=1}^{K} \frac{\partial \mathcal{L}_n^{\text{ZSL}}}{\partial y_i} x_j(\boldsymbol{a}_c) \tag{78}$$

Since the gradient has $K$ more terms and these updates are not independent from each other (since the final representations are used to construct a single logits vector after a dot-product with $\boldsymbol{z}_n$), this *may* lead to an increased overall gradient magnitude. We verify this empirically by computing the gradient magnitudes for our model and its non-ZSL "equivalent": a model with the same number of layers and hidden dimensionalities, but trained to classify objects in non-ZSL fashion.

To show that our class standardization procedure (9) smoothes the landscape, we apply Theorem 4.4 from Santurkar et al. (2018) that demonstrates that a model augmented with batch normalization (BN) has smaller Lipschitz constant. This is easily done after noticing that (9) is equivalent to BN, but without scaling/shifting and is applied in a class-wise instead of the batch-wise fashion.

We empirically validate the above observations on Figures 1 and 9.

### F.2 Formal reasoning

**ZSL embedders are prone to have more irregular loss surface**. We demonstrate that the loss surface of attribute embedder $P_{\boldsymbol{\theta}}$ is more irregular compared to a traditional neural network. The reason is that its output vectors $\boldsymbol{p}_c = P_{\boldsymbol{\theta}}(\boldsymbol{a}_c)$ are not later used independently, but instead combined together in a single matrix $W = [\boldsymbol{p}_1, ..., \boldsymbol{p}_{K_s}]$ to compute the logits vector $\boldsymbol{y} = W\boldsymbol{z}$. Because of this, the gradient update for $\theta_i$ receives $K$ signals instead of just 1 like for a traditional model, where $K$ is the number of classes.

Consider a classification neural network $F_{\boldsymbol{\psi}}(\boldsymbol{x})$ optimized with loss $\mathcal{L}$ and some its intermediate transformation $\boldsymbol{y} = W^\ell \boldsymbol{x}$. Then the gradient of $\mathcal{L}_n$ on $n$-th training example with respect to $W_{ij}^\ell$ is computed as:

$$\boldsymbol{y} = W^\ell \boldsymbol{x} \implies \frac{\partial \mathcal{L}_n}{\partial W_{ij}^\ell} = \frac{\partial \mathcal{L}_n}{\partial y_i^{(n)}} \frac{\partial y_i^{(n)}}{\partial W_{ij}^\ell} = \frac{\partial \mathcal{L}_n}{\partial y_i^{(n)}} x_j^{(n)} \tag{79}$$

While for attribute embedder $P_{\boldsymbol{\theta}}(\boldsymbol{a}_c)$, we have $K$ times more terms in the above sum since we perform $K$ forward passes for each individual class attribute vector $\boldsymbol{a}_c$. The gradient on $n$-th training example for its inner transformation $\boldsymbol{y} = W^\ell \boldsymbol{x}(\boldsymbol{a}_c)$ is computed as:

$$\boldsymbol{y} = W^\ell \boldsymbol{x}(\boldsymbol{a}_c) \implies \frac{\partial \mathcal{L}_n}{\partial W_{ij}^\ell} = \sum_{c=1}^{K} \frac{\partial \mathcal{L}_n}{\partial y_i} x_j(\boldsymbol{a}_c) \tag{80}$$

From this, we can see that the average gradient for $P_{\boldsymbol{\theta}}$ is $K$ times larger which may lead to the increased overall gradient magnitude and hence more irregular loss surface as defined in Section 3.5.

**CN smoothes the loss landscape**. In contrast to the previous point, we can prove this rigorously by applying Theorem 4.4 by Santurkar et al. (2018), who showed that performing standardization across hidden representations smoothes the loss surface of neural networks. Namely Santurkar et al. (2018) proved the following:

**Theorem 4.4 from (Santurkar et al., 2018)**. *For a network with BatchNorm with loss $\widehat{\mathcal{L}}$ and a network without BatchNorm with loss $\mathcal{L}$ if:*

$$g_\ell = \max_{\|X\| \leq \lambda} \|\nabla_W \mathcal{L}\|^2, \quad \hat{g}_\ell = \max_{\|X\| \leq \lambda} \left\|\nabla_W \widehat{\mathcal{L}}\right\|^2 \tag{81}$$

*then:*

$$\hat{g}_\ell \leq \frac{\gamma^2}{\sigma_\ell^2} \left( g_\ell^2 - m\mu_{g_\ell}^2 - \lambda^2 \langle \nabla_{\boldsymbol{y}_\ell} \mathcal{L}, \hat{\boldsymbol{y}}_\ell \rangle^2 \right) \tag{82}$$

where $\boldsymbol{y}_\ell, \hat{\boldsymbol{y}}_\ell$ are hidden representations at the $\ell$-th layer, $m$ is their dimensionality, $\sigma_\ell$ is their standard deviation, $\mu_g = \frac{1}{m}\left\langle \boldsymbol{1}, \partial\widehat{\mathcal{L}}/\partial\boldsymbol{z}_\ell \right\rangle$ for $\boldsymbol{z}_\ell = \gamma\boldsymbol{y}_\ell + \beta$, is the average gradient norm, $\gamma$ is the BN scaling parameter, $X$ is the input data matrix at layer $\ell$.

Now, it easy easy to see that our class standardization (9) is "equivalent" to BN (and thus the above theorem can be applied to our model):

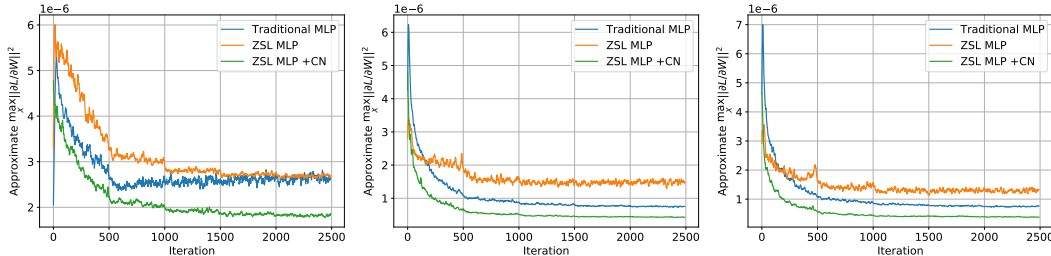

Figure 9: Empirical validation of the more irregular loss surface of ZSL models and smoothing effect of class normalization on other datasets. Like in figure 1, we observe that the gradient norms for traditional MLPs are much lower compared to a basic ZSL model, but class normalization partially remedies this problem.

- First, set $\gamma = 1$ (i.e. remove scaling) and $\beta = 0$ (i.e. remove bias addition).
- Second, apply this modified BN inside attribute embedder $P_{\boldsymbol{\theta}}$ on top of attributes representations $H = [\boldsymbol{h}_1, ..., \boldsymbol{h}_K]$ across $K$-axis (class dimension) instead of objects representations $X = [\boldsymbol{x}_1, ..., \boldsymbol{x}_B]$ across $B$-axis (batch dimension).

It is important to note here that there are no restricting assumptions on the loss function or what data $X$ is being used. Thus Theorem 4.4 of Santurkar et al. (2018) is applicable to our model which means that CN smoothes its loss surface.

### F.3 EMPIRICAL VALIDATION

To validate the above claim empirically, we approximate the quantity 11, but computed for all the parameters of the model instead of a single layer on each iteration. We do this by taking 10 random batches of size 256 from the dataset, adding $\boldsymbol{\epsilon} \sim \mathcal{N}(\mathbf{0}, I)$ noise to this batch, computing the gradient of the loss with respect to the parameters, then computing its norm scaled by $1/n$ factor to account for a small difference in number of parameters ($\approx 0.9$) between a ZSL model and a non-ZSL one. This approximates the quantity 11, but instead of approximating it around $\mathbf{0}$, we approximate it around real data points since it is more practically relevant. We run the described experiment for three models:

1. A vanilla MLP classifier, i.e. without any class attributes. For each dataset, it receives feature vector $\boldsymbol{z}$ and produces logits.
2. A vanilla MLP zero-shot classifier, as described in section 3.
3. An MLP zero-shot classifier with class normalization.

All three models were trained with cross-entropy loss with the same optimization hyperparameters: learning rate of 0.0001, batch size of 256, number of iterations of 2500. They had the same numbers of layers, which was equal to 3. The results are illustrated on figures 1 (left) and 9. As one can see, traditional MLP models indeed have more flat loss surface which is observed by a small gradient norm. But class normalization helps to reduce the gap.

## G CONTINUAL ZERO-SHOT LEARNING DETAILS

### G.1 CZSL EXPERIMENT DETAILS

As being said, we use the validation sequence approach from Chaudhry et al. (2019) to find the best hyperparameters for each method. We allocate the first 3 tasks to perform grid search over a fixed range. After the best hyperparameters have been found, we train the model from scratch for the rest of the tasks. The hyperparameter range for CZSL experiments are presented in Table 9 (we use the same range for all the experiments).

We train the model for 5 epochs on each task with SGD optimizer. We also found it beneficial to decrease learning rate after each task by a factor of 0.9. This is equivalent to using step-wise learning rate schedule with the number of epochs equal to the number of epochs per task. As being said, for CZSL experiments, we use an ImageNet-pretrained ResNet-18 model as our image encoder. In contrast with ZSL, we do not keep it fixed during training. The results for our mS, mU, mH, mJA, mAUC metrics, as well as the forgetting measure (Lopez-Paz & Ranzato, 2017) are presented on figures 10 and 11.

Table 9: Hyperparameters range for CZSL experiments

| | |
|---|---|
| Sampling distribution | uniform, normal |
| Gradient clipping value | 10, 100 |
| Attribute embedder learning rate | 0.001, 0.005 |
| Attribute embedder momentum | 0.9, 0.95 |
| Image encoder learning rate | 0.001, 0.005 |

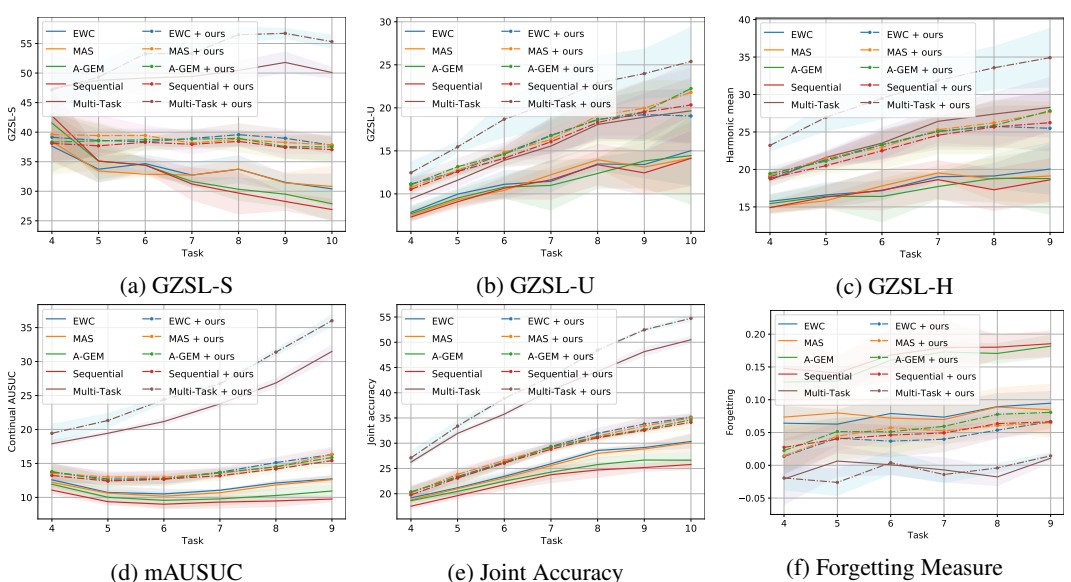

Figure 10: Additional CZSL results for CUB dataset

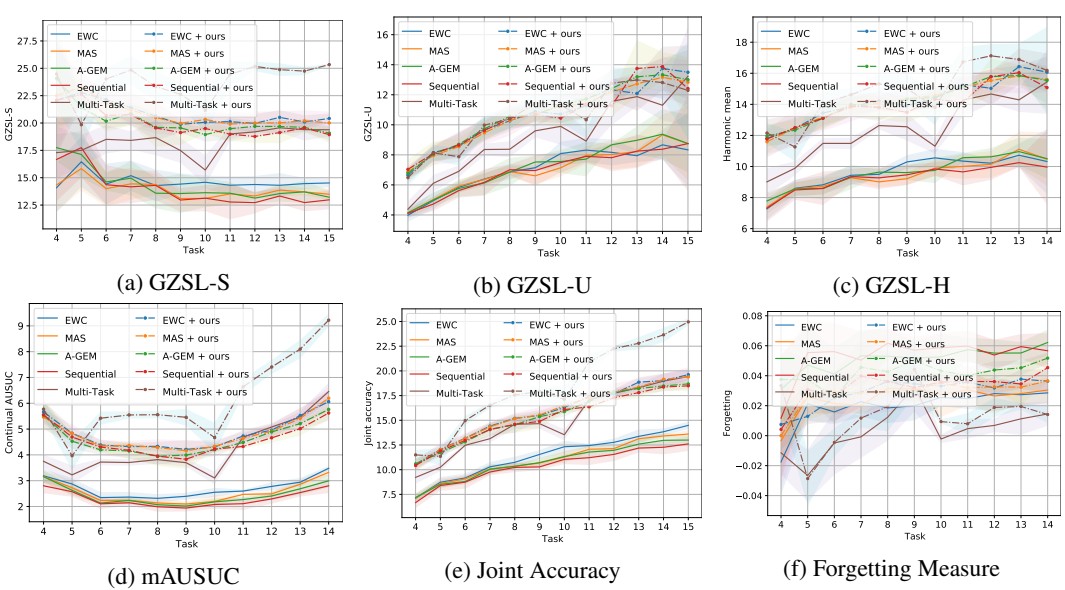

Figure 11: Additional CZSL results for SUN dataset

As one can clearly see, adding class normalization significantly improves the results, at some timesteps even surpussing the multi-task baseline without ClassNorm.

## G.2 Additional CZSL metrics

In this subsection, we describe our proposed CZSL metrics that are used to access a model's performance. Subscripts "tr"/"ts" denote train/test data.

**Mean Seen Accuracy (mSA)**. We compute GZSL-S after tasks $t = 1, .., T$ and take the average:

$$\text{mSA}(F) = \frac{1}{T} \sum_{t=1}^{T} \text{GZSL-S}(F, D_{\text{ts}}^{\leq t}, A^{\leq t}) \tag{83}$$

**Mean Unseen Accuracy (mUA)**. We compute GZSL-U after tasks $t = 1, ..., T - 1$ (we do not compute it after task $T$ since $D^{>T} = \varnothing$) and take the average:

$$\text{mUA}(F) = \frac{1}{T - 1} \sum_{t=1}^{T-1} \text{GZSL-U}(F, D_{\text{ts}}^{>t}, A^{>t}) \tag{84}$$

**Mean Harmonic Seen/Unseen Accuracy (mH)**. We compute GZSL-H after tasks $t = 1, ..., T - 1$ and take the average:

$$\text{mH}(F) = \frac{1}{T - 1} \sum_{t=1}^{T-1} \text{GZSL-H}(F, D_{\text{ts}}^{\leq t}, D_{\text{ts}}^{>t}, A) \tag{85}$$

**Mean Area Under Seen/Unseen Curve (mAUC)**. We compute AUSUC Chao et al. (2016) after tasks $t = 1, ..., T - 1$ and take the average:

$$\text{mAUC}(F) = \frac{1}{T - 1} \sum_{t=1}^{T-1} \text{AUSUC}(F, D_{\text{ts}}^{\leq t}, D_{\text{ts}}^{>t}, A) \tag{86}$$

AUSUC is a performance metric that allows to detect model's bias towards seen or unseen data and in our case it measures this in a continual fashion.

**Mean Joint Accuracy (mJA)**. On each task $t$ we compute the generalized accuracy on all the test data we have for the entire problem:

$$\text{mJA}(F) = \frac{1}{T} \sum_{t=1}^{T} \text{ACC}(F, D_{\text{ts}}, A) \tag{87}$$

This evaluation measure allows us to understand how far behind a model is from the traditional supervised classifiers. A perfect model would be able to generalize on all the unseen classes from the very first task and maintain the performance on par with normal classifiers.

# H Why cannot we have independence, zero-mean and same-variance assumptions for attributes in ZSL?

Usually, when deriving an initialization scheme, people assume that their random vectors have zero mean, the same coordinate variance and the coordinates are independent from each other. In the paper, we stated that these are unrealistic assumptions for class attributes in ZSL and in this section elaborate on it.

Attribute values for the common datasets need to be standardized to satisfy zero-mean and unit-variance (or any other same-variance) assumption. But it is not a sensible thing to do, if your data does not follow normal distribution, because it makes it likely to encounter a skewed long-tail distribution like the one illustrated on Figure 14. In reality, this does not break our theoretical derivations, but this creates an additional optimizational issue which hampers training and that we illustrate in Table 8. This observation is also confirmed by Changpinyo et al. (2016b).

If we do not use these assumptions but rather use zero-mean and unit-variance one (and enforce it during training), than the formula (6) will transform into:

$$\text{Var}[\tilde{y}_c] = d_z \cdot \text{Var}[z_i] \cdot \text{Var}[V_{ij}] \cdot \underset{a}{\mathbb{E}} [\|a\|_2^2] = d_z \cdot \text{Var}[z_i] \cdot \text{Var}[V_{ij}] \cdot d_a \tag{88}$$

Table 10: Checking how a model performs when we replace AN with the standardization procedure and with the standardization procedure, accounted for $1/d$ factor from (88). In the latter case, the performance is noticeably improved.

| | SUN | | | CUB | | | AwA1 | | | AwA2 | | |
|---|---|---|---|---|---|---|---|---|---|---|---|---|
| | U | S | H | U | S | H | U | S | H | U | S | H |
| Linear | 41.0 | 33.4 | 36.8 | 26.9 | 58.1 | 36.8 | 40.6 | 76.6 | 53.1 | 38.4 | 81.7 | 52.2 |
| Linear -AN | 13.8 | 41.0 | 20.6 | 16.8 | 62.0 | 26.4 | 16.8 | 74.2 | 27.4 | 18.9 | 73.2 | 30.0 |
| Linear -AN + $1/d_a$ | 36.2 | 33.0 | 34.5 | 36.0 | 39.0 | 37.4 | 49.2 | 72.3 | 58.6 | 46.9 | 79.5 | 59.0 |
| 2-layer MLP | 34.4 | 39.6 | 36.8 | 46.9 | 45.0 | 45.9 | 57.3 | 73.8 | 64.5 | 55.4 | 77.1 | 64.5 |
| 2-layer MLP -AN | 40.5 | 38.4 | 39.4 | 48.0 | 40.1 | 43.7 | 59.7 | 68.5 | 63.8 | 54.9 | 69.4 | 61.3 |
| 2-layer MLP -AN + $1/d_a$ | 37.1 | 38.4 | 37.7 | 50.8 | 33.3 | 40.2 | 60.1 | 66.3 | 63.1 | 50.5 | 71.8 | 59.3 |
| 3-layer MLP | 31.4 | 40.4 | 35.3 | 45.2 | 48.4 | 46.7 | 55.6 | 73.0 | 63.1 | 54.5 | 72.2 | 62.1 |
| 3-layer MLP -AN | 34.7 | 38.5 | 36.5 | 46.9 | 42.8 | 44.9 | 57.0 | 69.9 | 62.8 | 49.7 | 76.4 | 60.2 |
| 3-layer MLP -AN + $1/d_a$ | 42.0 | 33.4 | 37.2 | 50.4 | 30.6 | 38.1 | 57.1 | 64.7 | 60.6 | 55.2 | 69.0 | 61.4 |

(a) $\chi^2$-statistics of the normality test.

(b) Corresponding $p$-values

Figure 12: Results of the normality test for class attributes for real-world datasets. Higher values mean that the distribution is further away from a normal one. For a dataset of truly normal random variables, these values are usually in the range $[0, 5]$. As one can see from 12a, real-world distribution of attributes does not follow a normal one, thus requires more tackling and cannot be easily converted to it.

This means, that we need to adjust the initialization by a value $1/d_a$ to preserve the variance. This means, that we initialize the first projection matrix with the variance:

$$\text{Var}\left[V_{ij}\right] = \frac{1}{d_z} \cdot \frac{1}{d_a}. \tag{89}$$

In Table 10, we show what happens if we do count for this factor and if we don't. As one can see, just standardizing the attributes without accounting for $1/d_a$ factor leads to worse performance.

To show more rigorously that attributes do not follow normal distribution and are not independent from each other, we report two statistical results:

- Results of a normality test based on D'Agostino and Pearson's tests, which comes with scipy python stats library. We run it for each attribute dimension for each dataset and report the distribution of the resulted $\chi^2$-statistics with the corresponding $p$-values on Figure 12.

- Compute the distribution of absolute values of correlation coefficients between attribute dimensions. The results are presented on Figure 13 which demonstrates that attributes dimensions are not independent between each other in practice and thus we cannot use a common indepdence assumption when deriving the initialization sheme for a ZSL embedder.

We note, however, that attributes distribution is uni-modal and, in theory, it is possible to transform it into a normal one (by hacking it with log/inverse/sqrt/etc), but such an approach is far from being scalable. It is not scalable because transforming a non-normal distribution into a normal one is tricky and is done either manually by finding a proper transformation or by solving an optimization task. This is tedious to do for each dataset and thus scales poorly.

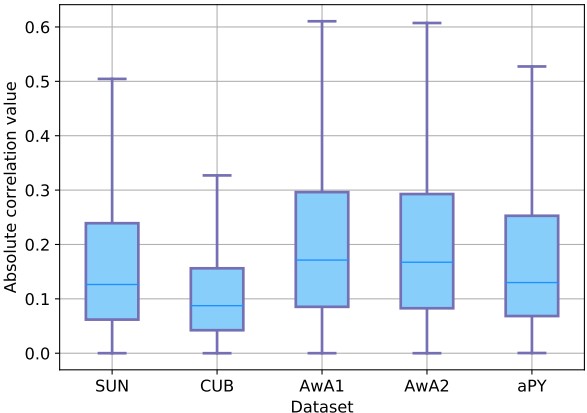

Figure 13: Distribution of mean absolute correlation values between different attribute dimensions. This figure shows that attributes are not independent that's why it would be unreasonable to use such an assumption. If attributes would be independent from each other, that would mean that, for example, that "having black stripes" is independent from "being orange", which tigers would argue not to be a natural assumption to make.

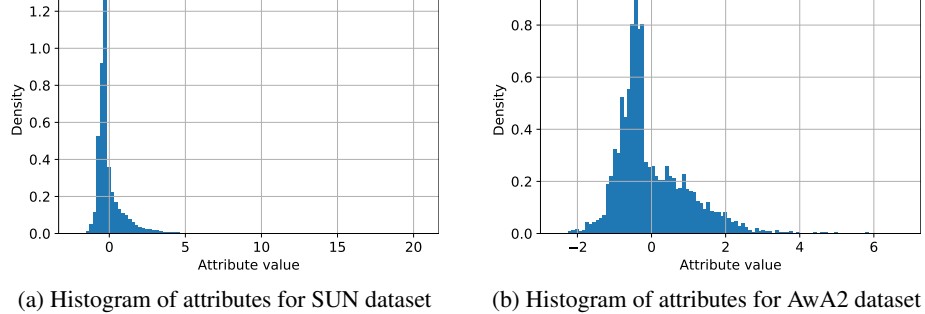

(a) Histogram of attributes for SUN dataset      (b) Histogram of attributes for AwA2 dataset

Figure 14: Histogram of standardized attribute values for SUN and AwA2. These figures demonstrate that the distribution is typically long-tailed and skewed, so it is far from being normal.

