# OpenReview forum: "Class Normalization for (Continual)? Generalized Zero-Shot Learning"
_ICLR.cc/2021/Conference — ICLR 2021 Poster_

### Official Review · AnonReviewer3 · 2020-10-25
**CLASS NORMALIZATION FOR ZERO-SHOT LEARNING**

**Rating:** 7
**Confidence:** 5

**Review:**

Summary: This paper presents a theoretical justification for normalization in model training on how it affects model performance and training time. It proposes two normalization tricks: normalize + scale trick and attributes normalization trick and apply in the zero-shot image classification task. This paper also shows that two normalization tricks are not enough to variance control in a deep architecture. To address this problem, a new initialization scheme is introduced. Apart from theoretical analysis and a new initialization scheme for normalization, it extends the zero-short learning approach in a continual learning framework. This new framework is called continual zero-shot learning (CZSL) and provides corresponding evaluation metrics. The experiments for CZSL are performed in two datasets, CUB and SUN.  This paper experimentally shows the effectiveness of the initialization, normalization, and scaling trick.

Strong Points:  1-  Paper is well organized and easy to follow.

2-  This paper took an interesting problem and developed a compelling investigation for how normalization affects performance. The theoretical justification for the normalization tricks sounds interesting and makes sense.

3-  It introduced two new techniques for normalization and shows by the theoretical justification that only normalization techniques are not sufficient for proper model training. For good model training apart from normalization tricks, it introduced an initialization scheme.

4- Using normalization tricks and a new initialization scheme reduces a significant model training time compared to previous approaches. It presents training speed results for several baseline approaches.

5- Innovative attempts in introducing a new ZSL problem, and several evaluation metrics are proposed for continual ZSL.


Weaknesses:  1- This paper presents a robust analysis of normalization, initialization, and scaling trick for ZSL. It also extends ZSL in continual learning.  I appreciate the author's effort for this solid analysis.  I expect a  new proposed model from authors to make the paper more strong.
2-  Missing comparison:  I recommend including paper [a] in the comparison table for CZSL.  Approach [a] is the first proposed baseline for continual zero-shot learning. Therefore it must be included in the comparison table.
3- Some recent state-of-the approaches are missing in the comparison table for ZSL. Please compare it with [b],[c] models.

4- Why have aPY dataset is not included in the experiments? Does this model not perform well in aPY dataset?
5- Is it possible for this normalization and scaling tricks for other applications such as object detection, action recognition, and image retrieval?
6-  I wonder by the training time you reported. I understand you have used quite a small neural network. Still, to have a clear view or fair comparison, you should have compared the timings with other initialization and default normalization and scaling tricks as well.

[a]- Lifelong Zero-Shot Learning, by Kun Wei et al. IJCAI 2020.
[b]- Episode-Based Prototype Generating Network for Zero-Shot Learning,  by  Yu et al. CVPR2020.
[c]- Meta-Learning for Generalized Zero-Shot Learning, by Verma et al. AAAI 2020.

Rating Reason: This paper has included a well-detailed analysis and mathematical formulations for normalization, initialization about model training. But it does not propose a novel model, which limits the novelty of the model.  CZSL formulation is also already explored in [a].

---

> ### Author Response · Authors · 2020-11-20
> **Authors' response to Reviewer 3 [Part 2]**
>
> > Is it possible for this normalization and scaling tricks for other applications such as object detection, action recognition, and image retrieval?
>
> We think class normalization can be applied to other domains with multiple modalities and where $y$ is a continuous variable as in zero-shot learning represented by class attributes or image-sentence retrieval. We chose to focus in this paper on the zero-shot learning and continual ZSL tasks, due to both their importance and clarity of evaluation.
>
> > You should have compared the timings with other initialization and default normalization and scaling tricks as well.
>
> Other initialization and normalization techniques would have precisely the same timings, but very low accuracy (see table 5). Our proposed model works that well because it follows a rigorously derived normalization scheme which adds negligible computation overhead. While modern ZSL methods also attain good performance — they achieve at much higher computational cost by using some sophisticated training, like GAN/VAE training with intricate loss terms or episode-based/meta-learning training schemes. This contrast is at the heart of our work: you do not need anything sophisticated to obtain SotA results, all you need is a good signal normalization.

---

> > ### Comment · AnonReviewer3 · 2020-11-21
> > **Convince with authors response**
> >
> > I appreciate to authors for their pointwise reply to each issue raised by reviewers. I have gone through other reviews and the author's response to them. I found an interesting thing that the authors did not try to escape from any reviewer's questions.
> >
> > The key advantage of this work is that it is the non-generative approach, which beats several generative approaches. More importantly, it is significantly efficient and fast compared to current state-of-the-art methods for zero-shot learning. I think it is because of the proposed normalization trick, and now it will be useful in zero-shot learning. As Reviewer R2 has suspected in the code and now verified it, he is convinced with the model performance and fastness. The authors have included the missing experiments and comparisons suggested by other reviewers and me in the revised version.
> >
> > Suggestion: Since this paper is mainly on zero-shot learning, I want to see a more realistic setting for continuous zero-shot learning. If each task consists of seen and unseen classes, it converges to an actual zero-shot setting with the same seen and unseen classes as the standard ZSL setup during the training of the last task.  Then compare this after last task training with ZSL approaches to see the degradation of model performance in task-wise sequential training. If possible author can include it in the ablation analysis.
> >
> > Overall: I found this approach is very interesting, simple with good performance concerning accuracy and execution time.  I hope the proposed normalization trick will help researchers to present more robust ZSL approaches. I tend towards acceptance from the borderline.

---

> ### Author Response · Authors · 2020-11-20
> **Authors' response to Reviewer 3 [Part 1]**
>
> We thank Reviewer3 for the valuable input and for appreciating our contribution. We address below the raised concerns.
>
> > This paper presents a robust analysis of normalization, initialization, and scaling trick for ZSL. It also extends ZSL in continual learning. I appreciate the author's effort for this solid analysis. I expect a new proposed model from authors to make the paper more strong.
>
> We believe that our paper includes multiple novel contributions. (1) we performed a novel theoretical analysis of normalization in ZSL models and validated it by extensive empirical evidence. (2) we proposed a novel class normalization technique that improves the scores considerably; (3) we also show that Class Normalization can improve performance on a continual version of zero-shot learning, CZSL, that we propose and contrasted it to related settings in the literature.
>
> > 2- Missing comparison: I recommend including paper [a] in the comparison table for CZSL. Approach [a] is the first proposed baseline for continual zero-shot learning. Therefore it must be included in the comparison
>
> 1. “Do we claim to be the first to study ZSL in continual setting?” No, but our CZSL setting is different as we explain. In our original submission, we cited and discussed [a] as Wei et al. (2020b)  but we think the first experiments on ZSL for continual learning was even earlier in (Arslan Chaudhry, Marc'aurelio Ranzato, et al.  ICLR,  2019). We developed CZSL as a generalized version of  (Arslan Chaudhry, Marc'aurelio Ranzato, et al.  ICLR,  2019) as we argued in the last paragraph in the Related work.
>
> 2. Difference between our new CZSL setting and [a]: CZSL is different from [a] and we think our CZSL benchmarks are both more challenging and more related to our goal. It is a development of the scenario proposed in Chaudhry et al. (2019), but authors there focused on ZSL performance only a single task ahead, while in our case we consider the performance on all seen (previous tasks) and all unseen data (future tasks). This definition also contrasts our work to the very recent work by Wei et al. (2020b), where a sequence of seen class splits of existing ZSL benchmarks is trained and the zero-shot performance is reported for every task individually at test time. In contrast, for our setup, the label space is not restricted and covers the spectrum of all previous tasks (seen tasks so far), and future tasks (unseen tasks so far). Due to this difference, we need to introduce a set of new metrics and benchmarks to measure this continual generalized ZSL skill over time.”
>
> 3. ”including paper [a] (i.e., Wei et al. (2020b)) in the comparison table”:  We tried to find the code/benchmarks of  Wei et al. (2020b) but were not able to find it. We think the current setup is more related to our work. We study the increasing ability of the model to recognize future tasks in a way that is distinguishable from seen classes till task $t$. We also think that the benchmarks we developed fulfills that purpose of showing that Class Norm is helpful for continual ZSL setting.
>
> > Please compare it with [b],[c] models.
>
> Thank you for the pointers! We already compare to EPGN [b] (See Table 2) and we will include the comparison to [c] for the updated version of the paper.
>
> > Why have aPY dataset not included in the experiments?
>
> The reason is that the very recent works tend not to benchmark on aPY. Examples from Table includes  LsrGAN (2020). DVBE (2020), EPGN (2020), TF-VAEGAN (2020), F-VAEGAN-D2 (2019). There is only a single method (DVBE) which computes the scores on aPY.  After taking a deeper look, we think reasons include that small number of classes, and the very sparse attribute representation across classes questioning if it needs to be improved and enriched. .
>
> aPY Experiment. We launched our method on aPY without any hyperparameters tuning whatsoever and obtained the score of 38.9. We believe that after tuning the hyperparameters, the scores could be improved. But even at the current state, this puts us on the third place after DVBE which has a score of 41.8 just right after CVC-ZSL with a score of 39.0 (very similar to our score). We prepared a [Google Colab notebook for aPY](https://gist.github.com/iclr2021-classnorm/cebb4dd7ca9ccc9a91d18b84aee056f7) that you can use to reproduce the results. We emphasize how simple our method is: all the code including data downloading/preprocessing/training/etc takes just ~150 lines of code. And for aPY, the training took just 8 seconds!

---

### Official Review · AnonReviewer4 · 2020-10-28
**An excellent paper investigating the normalization effect in the zero-shot learning context**

**Rating:** 8
**Confidence:** 5

**Review:**

================
Summary:

This paper provides a thorough analysis in the perspective of data variances on the widely used normalization tricks in the zero-shot learning research: normalize+scale and attribute normalization. It also demonstrates these tricks are not enough w.r.t. normalizing the variance in a non-linear model and propose a normalization trick to alleviate the issue. Both theoretical and empirical analysis are provided and results look convincing. Finally the authors propose a continual zero-shot learning problem scheme and illustrate some pioneering experimental results.

================
Reason for my score:

There is rare work on the normalization trick in the context of zero-shot learning, although techniques like attribute normalization are widely used in practice. This paper investigates the normalization effect extensively for zero-shot learning, and provides many insightful thoughts for utilizing these tricks. The authors also evaluate the proposed class normalization with a simple implementation on benchmark datasets and show convincing results. Such work makes good contributions to the related community and hence I give my score.

================
Pros:

1. The paper provides both theoretical and empirical analysis on the effect of commonly used normalization tricks for zero-shot learning, in the perspective of data variances.
2. The paper proposes a class normalization trick to alleviate the variance inflation/diminish in the non-linear model, and demonstrates its effectiveness on benchmark datasets.
3. The empirical analysis in the paper are extensive and convincing.
4. The paper also proposes a new framework of continual zero-shot learning.

================
Cons:

1. The paper didn't evaluate on another widely used benchmark dataset aPY, can author explain the reason?
2. On CUB dataset, the proposed method has a considerably large margin to the state-of-the-art methods, in contrast to other datasets. Is there any explanation on why it is the case? Have you tried to explain this failure especially from the perspective of the proposed normalization trick?

---

> ### Author Response · Authors · 2020-11-20
> **Authors' response to Reviewer 4**
>
> We thank Reviewer4 for the valuable input and for appreciating our contribution. We address below the raised concerns.
>
> > The paper didn't evaluate on another widely used benchmark dataset aPY, can author explain the reason?
>
> The reason is that the very recent works tend not to benchmark on aPY. Examples from Table includes  LsrGAN (2020). DVBE (2020), EPGN (2020), TF-VAEGAN (2020), F-VAEGAN-D2 (2019). There is only a single method (DVBE) which computes the scores on aPY.  After taking a deeper look, we think reasons 	include that small number of classes, and the very sparse attribute representation across classes questioning if it needs to be improved and enriched. .
>
> aPY Experiment. We launched our method on aPY without any hyperparameters tuning whatsoever and obtained the score of 38.9. We believe that after tuning the hyperparameters, the scores could be improved. But even at the current state, this puts us on the third place after DVBE which has a score of 41.8 just right after CVC-ZSL with a score of 39.0 (very similar to our score). We prepared a [Google Colab notebook for aPY](https://gist.github.com/iclr2021-classnorm/cebb4dd7ca9ccc9a91d18b84aee056f7) that you can use to reproduce the results. We emphasize how simple our method is: all the code including data downloading/preprocessing/training/etc takes just ~150 lines of code. And for aPY, the training took just 8 seconds!
>
> > On CUB dataset, the proposed method has a considerably large margin to the state-of-the-art methods, in contrast to other datasets. Is there any explanation on why it is the case? Have you tried to explain this failure especially from the perspective of the proposed normalization trick?
>
> We hypothesize that our method improves the scores by inherently allowing a model to better capture a signal from those attribute dimensions that are usually suppressed due to their low magnitude and by reducing the excessive influence of attributes with too high magnitudes. And for CUB, attributes are really well-balanced in terms of magnitudes compared to other datasets. We attach [histograms of averaged attributes magnitudes values](https://www.dropbox.com/s/xxo2mhdhl76q7rh/attributes-avg-magnitudes-hists.pdf?dl=0) for SUN, CUB, AwA and aPY. As you can see from these histograms, the attributes magnitude distribution for CUB has a much smaller tail, i.e. it is less “skewed” to the right. This makes the signal from large-magnitude attributes not suppress small-magnitude attributes as much as for other datasets. We achieve the best performance on AwA1/AwA2 and aPY datasets where the distribution is the most long-tailed (as we said, for aPY we achieved a GZSL-H score of 38.9 with the very first try without any bells and whistles).

---

> > ### Comment · AnonReviewer4 · 2020-11-22
> > **Response to authors update**
> >
> > Thanks for the authors' response. The results for aPY look also convincing and it is an interesting angle to connect the normalization to the attribute magnitude for the performance improvement. I opt to keep my score and recommend acceptance.

---

### Official Review · AnonReviewer2 · 2020-10-28
**Simple and effective approach**

**Rating:** 7
**Confidence:** 5

**Review:**

The paper shows that normalization is critical for zero-shot learning (ZSL). In the ZSL randomization is coming from the two sources, attribute and feature.  Normalization of the two source helps to reduce the variance. The paper uses an embedding based model where normalize visual feature are projected to the attribute space and in the attribute space, cosine similarity is measured to predict the class label. Paper also extend ZSL framework to the continual learning ZSL (CZSL) setup where whole data are not present at a time; instead, data comes in the form of a task, sequentially. The author shows that normalization helps to improve the CZSL result.

Positive:
1: The normalization is important in the NN/CNN model, in the ZSL, there are two sources of information, and proper normalization is important on both source for the better result. The paper gives a theoretical justification of why normalization is important and how we can do the same for the performance gain.

2: The proposed normalization shows the significant performance gain on the standard dataset for the GZSL setup (provided evaluation and code is correct).

3: Recently generative model shows the SOTA result for the GZSL setting since they can synthesize the unseen class samples and easily can handle the data biasness. It is nice to see that the non-generative model shows a significant improvement using the simple method, and it is much faster to train.

Comment:
1: The main concern is the result, I am unable to understand from where the exact gain is coming. Many previous works use Normalize+scale or normalization in the supervised learning or meta-learning scenario; it helps the generalization ability and smooth training and resulting in a performance gain. The performance gain using eq:[9] and [10] is expected, but [9]+[10] (CN) shows the much better result, I am unable to understand why this happens? I request the author; please explain the same.

2: The proposed approach is an embedding based model, it does not generate the samples from the unseen classes, then how model overcome the data biasness towards the seen class? Generally, it observes that seen-class shows the high accuracy and unseen-class show the low accuracy; hence H-mean is very poor. The generative model can handle this scenario since they can generate data from the unseen class. The normalization technique does not help to overcome the data biasness towards the seen-class then how model handle the data biases?

3: I appreciate the theoretical justification and identifying the problem in the deep model and providing the solution for that (section 3.4)

4: The comparison with the few recent meta-learning based approach [a] [b] [c] for the ZSL are missing, can you show the result compared with these approaches? Also, I request the author please provide the result if the same normalization is applied with the approach [c] (it is also embedding based model in the meta-learning framework).

[a] Episode-based Prototype Generating Network for Zero-Shot Learning, CVPR-20
[b] Meta-Learning for Generalized Zero-Shot Learning
[c] Learning to Compare: Relation Network for Few-Shot Learning, CVPR=18

Overall I like the idea and contribution, but I suspect the provided result for the GZSL result in table-2, I request the author please provide the code for the AWA2 and AWA1 dataset. I will further increase the score on the successful verification of the result.

4: If you don't use cosine similarity, then what is the dependency relation between the variance and weight W. I mean in the statement-1 if we don't use normalization then how variance depends on weight W? Also Var[\hat(y)_c] is independent of W, but learning is not independent of the initialization of W, in this case, variance does not matter, proper learning and generalization are more important.

5: I agree with the author that the provided evaluation metric for the CZSL is more generic and realistic. Here the model and CL procedure is not clear. What is Multi-task? What is sequential? How you ensure to overcome the catastrophic forgetting over the previous task while training the current task, The proper description is not provided, I request the author please provide the same.

6: In continual learning scenario with the increase of task, the model performance is degraded. It is shown in the figure-9, and 10 (supplementary) with the increase of the task model's performance is increasing, it means that model does not forget anything and also you have backward transfer how this is possible? Maybe I misunderstood something please explain.

---

> ### Author Response · Authors · 2020-11-20
> **Part 1: CN, Equation 9 and 10 and Adding [a,b,c] to the paper**
>
> We thank Reviewer2 for the valuable feedback. We here address the comments and will incorporate all the feedback.
>
>
> > The main concern is the result, I am unable to understand from where the exact gain is coming. Many previous works use Normalize+scale or normalization in the supervised learning or meta-learning scenario; it helps the generalization ability and smooth training and resulting in a performance gain. The performance gain using eq:[9] and [10] is expected, but [9]+[10] (CN) shows the much better result, I am unable to understand why this happens? I request the author; please explain the same.
>
> As you correctly noted, incorporating CN in its “full” form (i.e. using both eq 9 and eq 10) gives the best performance due to the better signal normalization and improved smoothness. The theoretical problem of using only eq 9 or eq 10 is that there is just no guarantee that this will lead to anything good. And from the practical perspective, we think that it should be beneficial to use eq 9 because standardization procedure smoothes the loss surface (see Appendix F and [1]). For example, Table 2 shows that just using eq 9 gives close performance as using eq 9 + eq 10. Eq 10 only does not perform well because without eq 9 it may make your signal vanish since what it does under the hood is just reducing the scale for the output matrix to make things align with the theory (which in turn leads to the better performance).
> - [1] https://arxiv.org/abs/1805.11604
>
> > The proposed approach is an embedding based model, it does not generate the samples from the unseen classes, then how does the model overcome the data biasness towards the seen class?
>
> Thanks for asking this interesting question. The “trick” is the following: previously (~before [1]), people trained a model which projects data onto the labels, i.e. a mapping X -> A from data space X to attribute space A. And this results in the very exact behaviour you foresee: the model becomes biased towards the seen representations which result in very low performance on the unseen. In our case, we follow the idea of [1, 2] and learn the mapping A->X. We describe the setting in Section 3 and depict the model on Figure 3. Table 2 of [2] shows how much difference this makes compared to using X->A projection. Since the model does not a direct access to seen data anymore, and its access to attributes happens only at the final discrimination phase, this avoids it getting biased towards the seen. Thank you for bringing this up, we will include this exposition in the updated version of the paper.
> - [1]Learning a Deep Embedding Model for Zero-Shot Learning Li Zhang, Tao Xiang, Shaogang Gong,  https://arxiv.org/abs/1611.05088
> - [2] Rethinking Zero-Shot Learning: A Conditional Visual Classification Perspective, https://arxiv.org/abs/1909.05995
>
> > The comparison with the few recent meta-learning based approach [a] [b] [c] for the ZSL are missing, can you show the result compared with these approaches?
>
> We already compare to [a] (see Table 2) and we will shortly include the missing comparisons in the nearest update.
>
> We follow up on the remaining concerns.

---

> > ### Comment · AnonReviewer2 · 2020-11-21
> > **Appreciate the response**
> >
> > Thanks to the authors for the detailed response. I highly appreciate the response and updated paper.
> >
> > The author answers all my query and provided the suggested comparison results. I have checked the provided code and able to reproduce the result.
> >
> > In the code, the author calibrates the seen class logits by a value 0.95 (hyperparameter) and it is not mentioned in the paper (logits[:, seen_mask] *= 0.95 # Trading a bit of gzsl-s for a bit of gzsl-u). Please mention this hyperparameter in the paper and explain how you obtained it. Also, include a small ablation over this value i.e. using the value [1.0,0.95,0.9,0.8,0.7,0.6,0.5] how results change. It will help to understand the sensitivity of this calibration parameter.
> >
> > The proposed model is simple and shows a significant improvement compared to recent baseline. Also, the proposed model is extremely fast to execute. The proposed metric for the continual ZSL is more realistic and the same normalization model helps to improve the continual learning baseline. I believe the proposed approach has good potential to improve the ZSL and continual ZSL framework, also in future, it will be interesting to see how the same normalization can be applied to the generative model and how much improvement we can obtain.
> >
> > Thanks

---

> > > ### Author Response · Authors · 2020-11-21
> > > **Response to Reviewer 2 update**
> > >
> > > Thanks, we are really grateful for your appreciation of our efforts.
> > >
> > > > In the code, the author calibrates the seen class logits by a value 0.95 (hyperparameter) and it is not mentioned in the paper. Please mention this hyperparameter in the paper and explain how you obtained it. Also, include a small ablation over this value.
> > >
> > > That technique is not original and was considered for example in [1,2]. Our difference to [1,2] is that we are multiplying by some value $s$ instead of adding some value $\tau$ to reweigh the logits since we think it to be more intuitive this way and the value of the weight now does not depend on the logits magnitudes. We find the optimal value of $s$ using cross-validation together with all the other hyperparameters.
> > >
> > > We added the discussion of this trick in Section 5 and the corresponding curves of how it influences GZSL-U/GZSL-S/GZSL-H scores in Appendix D.4 on Figure 4.
> > >
> > > > It will be interesting to see how the same normalization can be applied to the generative model and how much improvement we can obtain.
> > >
> > > We agree with this and we also think it is an exciting future direction. A key challenge here is that attribute embeddings are getting concatenated to a representation instead of being multiplied which should affect how the normalization occurs. Using other types of fusion like multiplicative interactions [3] should make the setup more similar to ours. Another challenge is that the optimization objective is very different (e.g., noise and adversarial losses), which may hide some subtleties that are important to consider while designing the normalization procedure.

---

> ### Author Response · Authors · 2020-11-20
> **Part2: Experiment: Adding Class Norm to Learning to Compare, CVPR18**
>
> > 4) Also, I request the author please provide the result if the same normalization is applied with the approach [c] (it is also embedding based models in the meta-learning framework).
>
> We did what you proposed and obtained the following results. First, we launched the [official source code as a baseline](https://github.com/lzrobots/LearningToCompare_ZSL) with the official hyperparameter just to see how it performs without our class normalization:
>
> AwA1:
> - ZSL-U: 0.6872959679507896
> - GZSL-U: 0.28563595890010673
> - GZSL-S: 0.8783777127463169
> - GZSL-H: 0.4310881673784809
>
> AwA2:
> - ZSL-U: 0.6517477335764157
> - GZSL-U:  0.10234362179045627
> - GZSL-S: 0.8809926564737267
> - GZSL-H: 0.1833838153382825
>
> CUB:
> - ZSL-U: 0.553778366712299
> - GZSL-U: 0.38336992978680834
> - GZSL-S: 0.6238278850778848
> - GZSL-H: 0.47489549514856744
>
> So, for some reason, the official baseline with the official hyperparameters diverged for us for AwA2 (the official reported result is 45.3) and we are not yet sure why. This much lower than expected performance is also observed in some of the [github issues](https://github.com/lzrobots/LearningToCompare_ZSL/issues/7).
> Then, we incorporated our class normalization to AttributeNetwork (which is a 2-layer MLP that embeds the attributes) and observed the following improvement in terms of scores:
>
> AwA1 + CN:
> - ZSL-U: 0.7104774542522156
> - GZSL-U: 0.29828608876422863
> - GZSL-S: 0.883896683187865
> - GZSL-H: 0.44604623033783647
>
> AwA2  + CN:
> - ZSL-U: 0.6519904295711082
> - GZSL-U: 0.12715849639164672
> - GZSL-S: 0.8884011086738732
> - GZSL-H: 0.2224738924395352
>
> CUB + CN:
> - ZSL-U: 0.5628699969293023
> - GZSL-U: 0.4006298170771589
> - GZSL-S: 0.6281890183721925
> - GZSL-H: 0.48924308702107
>
> We believe that the performance may be improved by selecting the hyperparameters that are different from the official ones but we have not done this exploration. We will incorporate these results into the paper after we’ll figure out why the official code diverges for AwA2 for both their provided setup and our class normalization.

---

> ### Author Response · Authors · 2020-11-20
> **Part 3: Collab/Code on AWA1 and AWA2**
>
> > Overall I like the idea and contribution, but I suspect the provided result for the GZSL result in table-2, I request the author please provide the code for the AWA2 and AWA1 dataset. I will further increase the score on the successful verification of the result.
>
> We prepared a minimal example of our approach as an [anonymized Google Colab Notebook](https://gist.github.com/iclr2021-classnorm/c29c8a1d4da78eb75a4cae24348b061d) that you can use to reproduce our results on AwA1/AwA2. It runs our method from the very scratch: from downloading the official data from the GBU website, preprocessing it, training and evaluating the model. We will release the full codebase of our project upon acceptance. We also revised the hyperparameters so they work better for the baselines. This decreases the gap between them to a value you observe for SUN/CUB datasets in Table 2.
>
> The implementation takes less than 150 lines of code including the code for data downloading, preprocessing, training, evaluation, and occasional comments. And the training time for it takes ~30 seconds! We hope that the simplicity of our approach is appreciated by the ML community.
>
> Also, after submission, we have found a mistake in our AWA1&2 ZSL evaluation procedure that made the model being evaluated in a class-balanced setting. However, AwA1 and AwA2 datasets are class-imbalanced (SUN and CUB are perfectly class-balanced and hence the results for them do not change). After fixing this error, our scores decreased noticeably but not dramatically. We still obtain SotA performance on AwA2 of GZSL-H = 67.6 with the closest runner up being DVBE with 67.0. Our results on AwA1 is 67.8, which is inferior to EPGN and CVC-ZSL having 71.2 and 69.1 respectively. Of course, we updated our results in the paper as well.
>
> Also, for your convenience, we launched the model for each setup of interest and saved the results as github gists (it just should be easier to browse things that way). They all differ only in the very first cell where we select the dataset and boolean flags for using/not using equation 9 or equation 10. Every other line of their code is the same:
> - [AWA1 (GZSL-H: 67.83)](https://gist.github.com/iclr2021-classnorm/c29c8a1d4da78eb75a4cae24348b061d)
> - [AWA1 without eq 9 (GZSL-H: 63.97)](https://gist.github.com/iclr2021-classnorm/22f8a40cd475898a7c25578a61c41810)
> - [AWA1 without eq 10 (GZSL-H: 65.94)](https://gist.github.com/iclr2021-classnorm/6a0abab160569573aebc54bc88c9dfef)
> - [AWA1 without eq 9 and eq 10 (GZSL-H: 62.77)](https://gist.github.com/iclr2021-classnorm/4f3c2d1a58d3e4321e545756908751a4)
>
> - [AWA2 (GZSL-H: 67.60)](https://gist.github.com/iclr2021-classnorm/dd0e89521be029dac63d1d8b0a2d9401)
> - [AWA2 without eq 9 (GZSL-H: 60.46)](https://gist.github.com/iclr2021-classnorm/d7e1723fbe6b2ed17c081b172e9c41b4)
> - [AWA2 without eq 10 (GZSL-H: 67.09)](https://gist.github.com/iclr2021-classnorm/5312dd0ccb84ae27e353213544720ad7)
> - [AWA2 without eq 9 and eq 10 (GZSL-H: 62.06)](https://gist.github.com/iclr2021-classnorm/c0a2d29c1f6b6eaed33392b930befe85)
>
>  In our case, the hardware was a Tesla P100 GPU (from `nvidia-smi`) and an Intel(R) Xeon(R) CPU @ 2.20GHz (from `cat /proc/cpuinfo`) which we recommend to reproduce our results. It may differ slightly if it was run on different hardware since that floating-point operation (summation/multiplication) may depend on each hardware.

---

> ### Author Response · Authors · 2020-11-20
> **Part4: response to other questions.**
>
> > 4: If you don't use cosine similarity, then what is the dependency relation between the variance and weight W. I mean in the statement-1 if we don't use normalization then how variance depends on weight W?
>
> If you do not use the normalize+scale, then the dependence would be the following (the derivation is similar to the derivation for other statements):
> $$
> y_c = z^\top Wh_c \Longrightarrow Var[y_c] = d_z^2 \cdot Var[z_i] \cdot Var[W_{ij}] \cdot E[\|\| h_c \|\|_2^2]
> $$
> So, it would depend linearly on the variance of $W$, multiplied by a very large constant $d_z^2$ (for our datasets of consideration, $d_z = 2048$).
>
> > 5: I agree with the author that the provided evaluation metric for the CZSL is more generic and realistic. Here the model and CL procedure is not clear. What is Multi-task? What is sequential?
>
> Multi-task model is a model that has access to all the previous data. In this way, it is an upper bound on the performance in CL. Sequential is a model that does not use any continual learning technique for its training and is used as a lower bound on the performance in CL. Thank you for the notice, we will add the description in the paper.
>
> > How you ensure to overcome the catastrophic forgetting over the previous task while training the current task, The proper description is not provided, I request the author please provide the same.
>
> CZSL provides you two ways to fight the catastrophic forgetting:
> 1. By incorporating CL techniques like EWC/MAS/A-GEM/etc that directly address the forgetting.
> 2. By incorporating ZSL techniques like our class normalization. And we believe that it is a much more interesting way. Because when improving your ZSL performance your model can in theory have “negative” forgetting (backward transfer), i.e. it can improve on a task after it learned it long time ago. It does so by learning more general attributes representations.
>
> > 6: In continual learning scenario with the increase of task, the model performance is degraded. It is shown in the figure-9, and 10 (supplementary) with the increase of the task model's performance is increasing, it means that model does not forget anything and also you have backward transfer how this is possible? Maybe I misunderstood something please explain.
>
> The main reason why our performance improves over time is because we consider the generalized setting, at each timestep we evaluate the model on all the tasks, including all the future ones. Imagine that we have 20 tasks to learn progressively on. After each task the model evaluated on all the 20 tasks. This means that at timestep 1 model’s objective is not easier that at timestep 20: it still had to solve all 20 tasks. This is a distinguishing feature of the generalized setting. In traditional continual learning, a model’s objective becomes harder and harder over time: after the first task, it should be able to solve well only the first task; after task 20 — it should be able to solve all 20 tasks. Thus in CL its performance decreases.

---

### Official Review · AnonReviewer1 · 2020-10-29
**Several major weaknesses**

**Rating:** 3
**Confidence:** 4

**Review:**

# Summary

This paper claims to have 3 main contributions.

C1: Understanding/Theory. It explains why the two tricks work in zero-shot learning (ZSL): (i) normalization + scaling in the compatibility function of the class features and the attributes, and (ii) attribute unit normalization.

C2: Method. It proposes a “class normalization” scheme (Eq. 9 and 10) and Fig. 3.
C2.1 From a “theoretical explanation” of C1 (ii), this fixes (ii) in a “deep” ZSL model.
C2.2 It improves “smoothness” of a “irregular” loss landscape in ZSL.

C3: Experiments. It demonstrates strong accuracy and training speed of the proposed approach in standard generalized ZSL. It also considers continual ZSL (Sect. 4), in which the proposed method is evaluated via mean accuracy (over timesteps) accuracy metrics and a forgetting metric.

###

# Strengths

S1. Simple method. This is a simple feature-attribute scoring function via scaled cosine similarity (with normalization).

S2. Strong empirical results (on both accuracy and training speed). See Table 2.

# Weaknesses

W1. Clarity

The organization of the paper is such that the reader has to refer to the appendix a lot. My biggest concern on clarity is on the “theoretical” results which are not rigorous and at times unsupported. Further, some statements/claims are not precise or clear enough for me to be convinced that the method is well-motivated and is doing what it is claimed to be doing.

W2. Soundness

I have a lot of concerns and questions here as I read through Sect. 3. At a high-level, I don’t see a clear connection between “improved variance control of prediction y^ or the smoothness of loss landscape” and “zero-shot learning effectiveness.” Details below. This is in part due to poor clarity.

W3. Experiments

IMO, if the main claim is really about the effectiveness of the two tricks and the proposed class normalization, then the experiments should go beyond one zero-shot learning starting point --- 3-layer MLP (Table 2).

- If baseline methods already adopt some of these tricks, it should be made clear and see if removing these tricks lead to inferior performance.
- If baseline methods do not adopt some of these tricks, these tricks, especially class normalization, could be applied to show improved performance. If it is difficult to apply these tricks, further explanation should be given (generally, also mention applicability of these tricks.)

This is done to some degree in the continual setting.

W4. Related work

As I mentioned in W3, it is unclear which methods are linear/deep, and which methods have already benefited from existing/proposed tricks.

###

# Detailed comments (mainly to clarify my points about weaknesses)


## Statement 1

The main claim for this part is that this statement provides “a theoretical understanding of the trick” and “allows to speed up the search [of the optimal value fo \gamma].”

However, I feel that we need further justifications on the correlation between Statement 1 (variance of y^_c, “better stability” and “the training would not stale”) and the zero-shot learning accuracy for this to be the “why normalization + scaling works.” My understanding is that the Appendix simply validates that Eq. (4) seems to hold in practice.

Moreover, is the usual search region [5,10] actually effective? Do we have stronger supporting empirical evidence than the three groups of practitioners (Li et al 2019, Zhang et al. 2019, Guo et al. 2020), who may have influenced each other, used it?

Finally, can the authors comment on the validity of multiple assumptions in Appendix A? To which degrees does each of them hold in practice?


## Statement 2 and 3

Why wouldn’t the following statement in Sect. 3.3 invalidate Statement 1?
“This may create an impression that it does not matter how we initialize the weights — normalization would undo any fluctuations. However it is not true, because it is still important how the signal flows, i.e. for an unnormalized and unscaled logit value”

It is unclear (at least not from the beginning) why understanding attribute normalization has to do with initialization of the weights.

Similar to my comments to Statement 1, why should we believe that the explanation in Sect. 3.3 and Sect. 3.4 is the reason for zero-shot learning effectiveness? In particular, the authors again claim that the main bottleneck in improving zero-shot learning is “variance control” (the end of Sect. 3.3).

I also have a hard time understanding some statements in Appendix H, which is needed to motivate the following statement in Sect. 3.3: “And these assumptions are safe to assume only for z but not for a_c, because they do not hold for the standard datasets (see Appendix H).”
H1: Would this statement still be true after we transform a_c with an MLP?
H2: Why is it not “a sensible thing to do” if we just want zero mean and unit variance?
H3: Why is “such an approach far from being scalable”?
H4: What if these are things like word embeddings?
H5: Fig. 12 and Fig. 13 are not explained.
H6: Histograms in Fig. 13 look quite normal.

How useful is Statement 2? Why is the connection with Xavier initialization important?

Why is “preserving the variance between z and y~” in Statement 3  important for zero-shot learning?


## Improved smoothness

The claim “improved smoothness” at the end of Sect. 3 and Appendix F is really hard to understand.
F1: How do the authors define “irregular loss surface”?
F2: “Santurkar et al. (2018) showed that batch-wise standardization procedure decreases the Lipschitz constant of a model, which suggests that our class-wise standardization will provide the same impact.” This is not very precise and seems unsupported. Please make it clear how. If this is a hypothesis, please make it clear.

Similarly to my comments to Statement 1-3, how is improved smoothness related to zero-shot learning effectiveness?


## Other more minor comments
1. Abstract: Are the authors the one to “generalize ZSL to a broader problem”? Please tone down the claim if not.
2. After Eq. (2): Why does attribute normalization look “inconsiderable” (possibly this is not the right word?) or why is it “surprising” that this is preferred in practice? Don’t most zero-shot learning methods use this (see for example Table 4 in [A])?
3. Suggestions for references for attribute normalization. This can be improved; I can trace this back to much earlier work such as [A] and [B] (though I think this fact is stated more explicitly in [A]).
4. Under Table 1 “These two tricks work well and normalize the variance to a unit value when the underlying ZSL model is linear (see Figure 1), but they fail when we use a multi-layer architecture.”: Could the authors provide a reference to evidence to support this? I think it is also important to provide a clear statement of what separates a “linear” or “multi-layer” model.
5. The first paragraph of Sect. 3: Could you provide references for motivations for different activation functions? Further, It is unclear that all of them perform normalization.
6. The second paragraph of Sect. 3: What exactly limits “the tools” for zero-shot learning vs. supervised learning? Further, it would also be nice to separate traditional supervised learning where classes are balanced and imbalanced; see, e.g., [C].
7. What is the closest existing zero-shot model to the one the authors describe in Sect. 3.1? Why is the described model considered/selected?

[A] Synthesized Classifiers for Zero-Shot Learning

[B] Zero-Shot Learning by Convex Combination of Semantic Embeddings

[C] Class-Balanced Loss Based on Effective Number of Samples

---

> ### Author Response · Authors · 2020-11-20
> **Part 1: possible misunderstandings on (Empirical Results & Theoretical Analysis)**
>
>
> We appreciate Reviewer1’s detailed and valuable feedback.  R1's review does not give credit to several efforts and key contributions in the paper, based on which we think there could be a misunderstanding. We hope it is alright that we highlight two key points that may help clarify our key message in the paper. We also acted with what is in our hands to improve the paper from an R1 perspective that we appreciated.
>
> ### Misunderstandings:
>
> **1) Our method effectiveness and Experimental Validation.**
> Apart from the theoretical analysis that we think is correct  and shows desirable learning characteristics of our study on normalization in ZSL, we have very solid empirical results and we believe that you didn’t give us enough credit for them:
>
> - Strong empirical results for ZSL on four datasets: our method is very simple and effective  (which we believe to be an important result), trains very fast ( about 1 minute), and beats modern SotA which usually follows a very complicated design and takes >1 hour to train.
> - We proposed a more rigorous formulation of CZSL as a direct generalization of ZSL + 5 novel metrics for it. We tested several continual learning benchmarks and showed that our method improved several baselines by a lot (~40% on average). We showed the value of CN on five CL methods (A-GEM, EWC, A-GEM, Sequential, and Multitask) on CUB and SUN datasets.
>
> **2)  Theoretical Analysis.**
>  We believe that the judgment used to evaluate our theoretical claims seems less related to our focus. We understood it as “ statement X shows that property Y holds, but this does not show that it improves ZSL performance. Thus, statement X is not sound/not rigorous.” While it is true that we do not prove the increased performance for ZSL directly, we think this does not make our theoretical analysis not sound as we detail below:
>
> - For each statement, we provided very solid empirical evidence and showed that it correlates with the good ZSL performance in practice. For example, our proposed model has more steady and closer-to-unit variance (shown by statement 3 and figures 1,5,6,7,8) while other methods do not — and it also outperforms them in terms of performance. Or it has a smoother loss landscape, shown from the exposition in Appendix F.
>
> - The current state of deep learning theory is at such a level that proving that some property would directly increase the performance is almost always out of reach unless you have a very specific simplified model (like infinite-width networks) trained for a very well-studied task like binary classification where you have a lot of existing theoretical tricks to employ. Let's consider for example the task we are solving: investigating the analysis of variance inside a network. One of the most famous early works on it is Xavier init [1]. In their paper, authors derive a proper initialization scale for neural network weights that would preserve the variance during a forward or backward pass — an analysis which is similar in spirit to what we did for ZSL models. Then they provide strong empirical evidence of why it is a good thing (which we also do!) by showing that the theoretical claims hold in practice and correlates with the improved performance. But they do not have any proof that the variance preservation directly improves a model’s performance — would you discard their contributions as well? And we still do not have a rigorous theoretical connection between this “variance preservation” and final model performance. Why? Because it is too hard. There is a series of works on dynamical isometry [2,3,4] that took the authors years to develop, and we still do not have an exact connection between the signal propagation and the model scores! In their works, they show the “trainability” of a model depending on the initialization but this “trainability” is not directly connected to the performance, but rather as a property of the Jacobian matrix — which, we emphasize it again — is not directly connected to the score.
>
> With all this being said, we disagree with how you evaluate theoretical analysis. And we believe that it is a decent way to develop the understanding of deep learning the way it is done in the paper: first, formulate and prove a statement that some property holds; and second, demonstrate rigorous evidence that supports both the statement and how it correlates with the performance.
>
> - [1] http://proceedings.mlr.press/v9/glorot10a/glorot10a.pdf
> - [2] https://arxiv.org/abs/1711.04735
> - [3] https://arxiv.org/abs/1806.05393
> - [4] https://arxiv.org/abs/1901.08987

---

> ### Author Response · Authors · 2020-11-20
> **Part 2: responses to minor concerns W1-W4**
>
> ### Detailed response
> > W1: The organization of the paper is such that the reader has to refer to the appendix a lot.
>
> We think this may relate to the misunderstanding we clarified in part 1 and hence we think it is reasonable that we structured the paper in the current form. Based on part1 clarification, we include informal versions of the statements in the main paper and the appendix contains only the fine details for interested readers (assumptions, rigorous formulations, and the proofs). The purpose of the informal statements is to provide the key observations that motivate our techniques. Providing informal statements in the main body and putting the details in the appendix is common (examples [1,2,3,4]). Our main goal is to show the impact of normalization and to understand its effectiveness and we hope that our paper may encourage future work to develop more understanding on top of that.
> - [1] (Jaehoon Lee, etl, NeurIPS, 2019). https://arxiv.org/abs/1902.06720
> - [2] (Qianxiao Li, Shuji Hao etal, 2018 ),  https://arxiv.org/abs/1803.01299
> - [3](Uri Shaham, et,al, 2015)  https://arxiv.org/abs/1509.07385
> - [4] (William H. Guss, et,al,2018 ), https://arxiv.org/abs/1802.04443
>
> >  W1: My biggest concern on clarity is on the “theoretical” results which are not rigorous and at times unsupported.
>
> Could you please elaborate on what you mean by “not rigorous and at times unsupported”? If you found a mistake in any of the proofs, we would appreciate it if you’ll point it out. Could you please be specific which theoretical result is “unsupported” and where?
>
> > W1:  Further, some statements/claims are not precise or clear enough for me to be convinced that the method is well-motivated and is doing what it is claimed to be doing.
>
> We provided extensive empirical validation of our claims and presented it on Figures 1 in the main body of the paper and in appendices A, B, E and F. They clearly demonstrate that those benchmarks which tend to have more steady close-to-unit variance for logits/pre-logits activations — also tend to perform better in practice also (see Table 2 and Table 4). This gives us the evidence to state that “good” variance is indeed a good thing. Our class normalization scheme is directly motivated by the theoretical statements: Statement 2 shows the effect of attributes normalization on a linear embedder. And if you now replace a linear embedder with the non-linear one, then to show that the normalization is now lost, you just need to derive the variance formula using the proof Statement 2 but replacing letter “a” with the letter “h” everywhere as we state in Section 3 and Appendix B.
> > If the main claim is really about the effectiveness of the two tricks and the proposed class normalization, then the experiments should go beyond one zero-shot learning starting point --- 3-layer MLP (Table 2)
>
> A 3-layer MLP is a good “working horse” since it is much simpler than any other ZSL method and it illustrates our point the best. If we would build upon an existing method, that would only obscure things. And since we achieve state-of-the-art performance with a 3-layer MLP — this only strengthens our point. We believe that our experiments in full demonstrate the effectiveness of the two tricks and the proposed class normalization in zero-shot learning scenario.
>
> > W2
>
>  please see in part 1
>
> > W3: Also mention applicability of these tricks
>
> We thank R1 and we appreciate raising this reasonable concern. Our method is applied in those models which use structured embeddings as an additional input source: zero-shot learning, metric learning, image retrieval, etc. We are adding the discussion to the updated version of the paper and leave the experiments in other areas for future exploration.
>
> >W4:  [Related work] As I mentioned in W3, it is unclear which methods are linear/deep, and which methods have already benefited from existing/proposed tricks.
>
> Linear models were “popular” in the early years of ZSL [1, 2, 3] and recent methods employ deep embedding-based ZSL models ([4, 5, 6, 7]). Normalizing attributes became the norm after [8] and normalize+scale tricks found its application in ZSL in [7,4] and their continuations.
> - [1] http://proceedings.mlr.press/v37/romera-paredes15.pdf
> - [2] https://proceedings.neurips.cc/paper/2007/file/ed265bc903a5a097f61d3ec064d96d2e-Paper.pdf
> - [3] https://arxiv.org/abs/1409.8403
> - [4] https://arxiv.org/abs/1909.05995
> - [5] https://arxiv.org/abs/1711.06025
> - [6] https://arxiv.org/abs/1611.05088
> - [7] https://papers.nips.cc/paper/2018/hash/1587965fb4d4b5afe8428a4a024feb0d-Abstract.html
> - [8] https://arxiv.org/abs/1703.04394

---

> ### Author Response · Authors · 2020-11-20
> **Part 3:  Statement 1**
>
> >I feel that we need further justifications on the correlation between Statement 1 (variance of y^_c, “better stability” and “the training would not stale”) and the zero-shot learning accuracy for this to be the “why normalization + scaling works.” My understanding is that the Appendix simply validates that Eq. (4) seems to hold in practice.
>
> Statement 1 derives the formula for the logits variance when the NS trick is applied and we show in Figures 1,4,5,6, and 7 that the NS trick indeed improves it in practice when the proper $\gamma$ is used. Tables 2 and 4 show the performance for model +NS and models for different \gamma values and demonstrate that using the proper value of \gamma favourable influences performance. We believe that this is enough correlation to state that the proper variance correlates with the good performance.
>
> > Moreover, is the usual search region [5,10] actually effective? Do we have stronger supporting empirical evidence than the three groups of practitioners (Li et al 2019, Zhang et al. 2019, Guo et al. 2020), who may have influenced each other, used it?
>
> We provided several works that select their final values from this region: [Li et al. 2019] — uses \gamma=10, [Guo et al. 2020] — uses \gamma=8, [Zhang et al. 2019] — uses \gamma=3 but their embedding size is much smaller (since we normalize by smaller norm, we need smaller scaling afterwards), [1] — uses \gamma=8, [2] — uses $\gamma=10$, [3] set it to 10 and optimized during training. They indeed may have influenced each other, but tracking down what exact hyperparameter search regions they used is out of reach and we can report only their final values.
>
> - [1] https://arxiv.org/abs/1801.07698
> - [2] https://papers.nips.cc/paper/2018/hash/1587965fb4d4b5afe8428a4a024feb0d-Abstract.html
> - [3] Dynamic few-shot visual learning without forgetting, CVPR 2018
>
> > Finally, can the authors comment on the validity of multiple assumptions in Appendix A? To which degrees does each of them hold in practice?
>
> Our assumptions 1,3,4,5 are typical for such kind of analysis and Xavier init [1], Kaiming init [2], Hyper init [3] all use them as well. Assumption 2 is a technical assumption that we just need to apply the CLT. Assumption 6, as noted in the paper, holds only for large-dimensional inputs, but this is exactly our case. Figure 2 in Appendix A demonstrates that these assumptions lead to a pretty good approximation. We will include the discussion in the paper.
> - [1] http://proceedings.mlr.press/v9/glorot10a/glorot10a.pdf
> - [2] https://arxiv.org/abs/1502.01852
> - [3] https://openreview.net/forum?id=H1lma24tPB

---

> ### Author Response · Authors · 2020-11-20
> **Part 4: Statement 2-3**
>
>
> > Why wouldn’t the following statement in Sect. 3.3 invalidate Statement 1? “This may create an impression that it does not matter how we initialize the weights — normalization would undo any fluctuations. However it is not true, because it is still important how the signal flows, i.e. for an unnormalized and unscaled logit value”
>
> Statement 1 operates on top of logits, while Statement 2 operates on top of pre-logits. Statement 1 makes a claim about the forward pass and has nothing to do with the backward pass, while Statement 2 shows that incorporating attributes normalization is equivalent to employing Xavier fan-out init, which adjusts the variance of the backward pass.
>
> > It is unclear (at least not from the beginning) why understanding attribute normalization has to do with initialization of the weights.
>
> Dividing the attributes by some large value (in this case, by their norm) is equivalent to dividing the subsequent dense layer weight matrix by this large value. Hence this is equivalent to reparametrizing the weight matrix with the different scale. Thank you for pointing this out, we will clarify that moment in the updated version of the paper.
>
> > Similar to my comments to Statement 1, why should we believe that the explanation in Sect. 3.3 and Sect. 3.4 is the reason for zero-shot learning effectiveness?
>
> We provide a large empirical investigation (Figures 1, 4, 5, 6, 7) on how the variance behaves for different setups and how they perform in practice, thus obtaining the evidence on the correlation.
>
> > The authors again claim that the main bottleneck in improving zero-shot learning is “variance control” (the end of Sect. 3.3).
>
> Though we didn’t claim such a thing (we claimed that Xavier/Kaiming init would use invalid assumptions for the attributes thus producing an improper scaling for weight initialization which in turn would translate into the bad variance control), our empirical results demonstrate that after adjusting it, you obtain strong state-of-the-art performance.
>
> > I also have a hard time understanding some statements in Appendix H, which is needed to motivate the following statement in Sect. 3.3: “And these assumptions are safe to assume only for z but not for a_c, because they do not hold for the standard datasets (see Appendix H).” H1: Would this statement still be true after we transform a_c with an MLP? H2: Why is it not “a sensible thing to do” if we just want zero mean and unit variance? H3: Why is “such an approach far from being scalable”? H4: What if these are things like word embeddings? H5: Fig. 12 and Fig. 13 are not explained. H6: Histograms in Fig. 13 look quite normal.
>
> - H1: Assumptions would (approximately) start working after we transform a_c with an MLP, but if we’ll apply the MLP without using eq 9 + eq 10 this will lead to a “bad” variance (as we demonstrate on figures 1, 4, 5, 6, 7).
> - H2: It just works worse in practice. Typically, things work the best when your data follows N(0,1). Attributes do not follow normal distribution originally and thus cannot be converted to N(0,1) just by standardization.
> - H3: Because there is no magic formula of transforming a non-normal distribution to a normal one. Especially multi-modals. You should either solve a tedious optimization task or do this manually. This is not scalable when you are solving task by task.
> - H4: It depends on the exact situation, but we believe that the coordinates in word embeddings in general follow the normal distribution.
> - H5: We missed to add references from text to the figures (thanks for pointing this out, we’ll fix this), but they have explanatory captions which should not confuse a reader too much.
> - H6: In our view, histograms in Fig. 13 look is not normal for the following reasons: 1) the long tail on the right showing that the distribution is obviously skewed. (2) To remove any bias from deciding what looks normal and what is not, we incorporated D’Agostino and Pearson’s normality tests resulting in Figure 11, which shows that attributes are distributed very far from normal.
>
> > How useful is Statement 2? Why is the connection with Xavier initialization important?
>
> Statement 2 shows that when you use attribute normalization in a linear embedding-based setting, then your init is equivalent to Xavier fan-out init. It is just a coincidence that it turned out to be equivalent to Xavier fan-out one. Xavier fan-out init is a decent, but not the best choice for the model initialization. A more popular way toinitialize linear layers is via Kaiming fan-in init with the uniform distribution — a [default init for pytorch nn.Linear](https://github.com/pytorch/pytorch/blob/master/torch/nn/modules/linear.py#L87).

---

> ### Author Response · Authors · 2020-11-20
> **Part 5: Statement 3 and other minor comments**
>
> > Why is “preserving the variance between z and y~” in Statement 3 important for zero-shot learning?
>
> Preserving the variance was shown to be effective in classical supervised learning, demonstrated by Xavier/Kaiming inits. It make the signal propagation independent of the width (at least asymptotically) and making us to more likely expect good gradients in return. This motivates us to study the variance effect in ZSL task to see if it provides performance gains in this case. The precise, direct way of how the variance changes the final model performance is unknown even for the supervised learning regime.
>
> > Abstract: Are the authors the one to “generalize ZSL to a broader problem”? Please tone down the claim if not.
>
> Our CZSL is formulated in such a way that ZSL is a specific case of it when there are only 2 tasks, where the first task has the seen classes and the second task has the unseen classes. Thus CZSL is a generalization of ZSL.
>
> > After Eq. (2): Why does attribute normalization look “inconsiderable” (possibly this is not the right word?) or why is it “surprising” that this is preferred in practice? Don’t most zero-shot learning methods use this (see for example Table 4 in [A])?
>
> A common practice is to normalize your data to zero-mean and unit-variance and here we normalize attributes to a unit norm instead which diverges with that common practice. Thus we found this to be surprising.
>
> > Suggestions for references for attribute normalization. This can be improved; I can trace this back to much earlier work such as [A] and [B] (though I think this fact is stated more explicitly in [A]).
>
> Thanks, we will incorporate this in the revised version of the paper.
>
> > Under Table 1 “These two tricks work well and normalize the variance to a unit value when the underlying ZSL model is linear (see Figure 1), but they fail when we use a multi-layer architecture.”: Could the authors provide a reference to evidence to support this? I think it is also important to provide a clear statement of what separates a “linear” or “multi-layer” model.
>
> Figure 1 (two left plots) and Figures 4,5,6,7 illustrate the variances for different setups. In particular, you may see that while the variance for a linear model +NS+AN is close to unit, the variance for the non-linear model +NS+AN noticeably reduces below 1. Adding CN improves the situation. The difference between a linear and a multi-layer model is in the amount of layers being used. For a linear model we use just a single layer, for a multi-layer model we use 2 additional layers.
>
> > The first paragraph of Sect. 3: Could you provide references for motivations for different activation functions? Further, It is unclear that all of them perform normalization.
>
> Could you please elaborate on your request? It is true that not all activation functions perform normalization, but some of them do, like SELU [1]. However, it is not clear how it is related to the first paragraph of Section 3. Motivations for different activation functions are found in papers that introduce them.
> - [1] https://arxiv.org/abs/1706.02515
>
> > The second paragraph of Sect. 3: What exactly limits “the tools” for zero-shot learning vs. supervised learning? Further, it would also be nice to separate traditional supervised learning where classes are balanced and imbalanced; see, e.g., [C].
>
> First, the task is different and you have to consider two different sources of inputs. This introduces some “fusion” step between the data and the attributes which complicates things. Second, attributes follow a different distribution compared to what one would assume for “traditional” data.
>
> > What is the closest existing zero-shot model to the one the authors describe in Sect. 3.1? Why is the described model considered/selected?
>
> The closest model is an embedding-based model from [https://arxiv.org/abs/1909.05995], but we do not use their sophisticated episode-based training scheme. We build upon it since it is much simpler (when you remove the episode-based training scheme) and provides decent performance (see Table 1 and Table 2 of our paper).

---

> ### Author Response · Authors · 2020-11-22
> **Additional paper update based on Reviewer 1 feedback**
>
> We want one more time to thank Reviewer 1 for their diligent feedback, and we did what was within our hands to improve the paper based upon it. Among other things, this includes the following improvements:
>
> - We additionally applied our model's idea to RelationNet and CVC-ZSL methods and observed the improvement in both cases by +2.0 and +1.8 on average in terms of GZSL-H for them, respectively. We report these results in Appendix D.5 of the paper.
> - We enriched the exposition on attributes normalization, normalize+scale, and embedding-based ZSL models in the Related Work section (first two paragraphs).
> - We improved our paper's clarity for the smoothness and normalization exposition and added additional details for them (sections 3.3-3.4 and Appendix F).
> - We added the discussion of figures 12-13 in Appendix H
> - We provided additional discussion on how realistic the assumptions are in Appendix A
> - Launched additional experiments to demonstrate the importance of AN/NS for other models: 1/2-layer MLP and CVC-ZSL (see Table 8 in Appendix D.6)
> - Added exposition with the experimental validation of what happens when we manually enforce zero-mean/unit-variance attributes assumptions instead of using the formula derived with their relaxation in Appendix H
> - Rewrote the exposition on the loss surface smoothness and moved it into a separation Section 3.5
>
> We respectfully disagree with Reviewer 1 judgment of our theoretical contributions as unsound/unrigorous and find this feedback not being specific enough to give us a chance to act upon. Our provided empirical evidence is in line with Xavier init [1], Kaiming init [2], Hyper init [3] works, which we consider to be sound, consistent, and rigorous.
>
> Finally, we believe that Reviewer 1 rating didn't give our work credit for the practical contributions in terms of the novel, simple, and highly efficient ZSL algorithm and the novel exposition of CZSL with 5 new evaluation criteria and strong practical results.
>
> - [1] http://proceedings.mlr.press/v9/glorot10a/glorot10a.pdf
> - [2] https://arxiv.org/abs/1502.01852
> - [3] https://openreview.net/forum?id=H1lma24tPB

---

> ### Comment · AnonReviewer1 · 2020-11-23
> **Thank you for your feedback**
>
> I'd like to thank the authors for trying to address my concerns in detail. I'd like to first respond to your major point, which I suspect we don't agree, but I'd be happy to take in further feedback from the authors and other reviewers.
>
> In order to judge the significance of the contribution "showing that Statement X has Property Y", I think one could use (i) the difficulty in showing that Statement X has Property Y in theory, (ii) how much does "Statement X has Property Y" hold in practice (see property Y in experiments with real datasets, reasonable assumptions?), and (iii) Property Y helps downstream Z. My understanding is that this paper doesn't claim (i) in contrast to, say, theory papers that involve a lot of technical and mental gymnastics.
>
> Now, (iii) Y and Z (zero-shot accuracy) are weakly correlated in this paper. IMO, "extensive evaluation" in Figures 1, 4, 5, 6, 7 and the tables shows (ii) but provides only weak empirical evidence to support (iii) (the authors disagree). Further, the tricks have been shown to work only on one zero-shot learning MLP baseline in this paper (the authors believe this is enough). How should we deduct the generalizability of this correlation? If this bar is unreasonable, I'd be willing to increase my score.
>
> Another major criticism is also with respect to the clarity (which must reach a level to support soundness/rigorousness), and I understand that this may not be related to proving (iii). I tried to ask questions and gave a lot of examples that I feel are enough to illustrate this point. For example, see F1, F2, H2. Note that my concerns in H also weaken (ii) (see also the authors' responses to H1, H2, H4, H6).

---

> > ### Author Response · Authors · 2020-11-25
> > **Additional response for Reviewer 1's concerns on F1, F2 and H1, H2, H4, H6 including updates (Sec 3.5 and results)**
> >
> > > Another major criticism is also with respect to the clarity [...] For example, see F1, F2, H2.
> >
> > We provided additional details about F1, F2 and other Reviewer 1’s concerns in our [previous update](https://openreview.net/forum?id=7pgFL2Dkyyy&noteId=1UGLzHUd3vl) and rewritten this section and elaborated on H in the current one. If Reviewer 1 believes that there are other places to improve the clarity in, we would appreciate the pointers.
> >
> > > F1: How do the authors define “irregular loss surface”?
> >
> > We borrow the definition of the loss surface smoothness from [4] and define it via Lipshitzness (see the newly added Section 3.5).
> >
> > > F2: [Our statement claiming that CN smoothes the surface] This is not very precise and seems unsupported. Please make it clear how. If this is a hypothesis, please make it clear.
> >
> > We greatly elaborated on this point in Section 3.5 and Appendix F. To show that CN smoothes the loss surface, we first notice that the class standardization (eq 6) is equivalent to batch normalization with 1) scaling/bias removed and 2) applied across class-dimension on top of attribute representations. This makes it straightforward to apply Theorem 4.4 from [4] (see Appendix F). Hence, this claim holds rigorously, in contrast to our other claim about more irregular loss surface of ZSL embedders which we state only as a hypothesis (but also validate empirically). See Section 3.5 for the details.
> >
> > > Note that my concerns in H also weaken (ii) (see also the authors' responses to H1, H2, H4, H6).
> >
> > We respectfully disagree and we believe they do not. We elaborate here on our answers on H1, H2, H4 and H6 and provide an additional context since we believe that our previous response on them produced a misunderstanding. As we understood it, Reviewer 1 is concerned about our claim that zero-mean and unit-variance assumptions do not apply to attribute vectors in ZSL, which motivated us to relax these assumptions for Statement 2. Reviewer 1’s point is that one can enforce these assumptions by manually standardizing attributes to zero-mean and unit variance.
> >
> > > H1: Would this statement still be true after we transform a_c with an MLP?
> >
> > The MLP outputs would have zero-mean and some variance  (maybe unit — depending on the initialization), but here we are delegating the job of proper initialization with respect to this MLP component instead of with respect to the attributes. And this defines our class normalization purpose which is about preserving the variance when a deep MLP is employed to transform the attributes.
> >
> > > H2: Why is [standardization] not “a sensible thing to do” if we just want zero mean and unit variance?
> >
> > Manually standardizing the attributes would simplify the variance formula (6) but at the expense of deviating from the common practice [1,2] as we elaborated above. This would slightly change this formula (6) and now instead of multiplication by $E[\|a\|_2^2]$, we will have the multiplication by $d_a$ (attributes dimensionality) — we show this in Appendix H. What is important here is that we still cannot employ default Kaiming/Xavier scalings since they do not account for this additional $d_a$ factor.
> >
> > Additional Experiments. To demonstrate this, we ran additional experiments and added the results that describe this case in Table 10: if one standardizes attributes to zero-mean and unit-variance instead of the normalization and employs the default (Kaiming) init, then this worsens the scores. But if we adjust for this transformation with the appropriate $1/d_a$ factor (equation 90 in Appendix H), then the performance restores (and even outperforms somewhere). The factor $1/d_a$ is derived as a variance-preserving factor (eq. 90), and since it resulted in a big influence on the scores (~20% improvement — see Table 10), it suggests that our hypothesis that improving the variance improves the performance is trustworthy.
> >
> > > H4: What if these are things like word embeddings?
> >
> > As we said in our previous response, we believe that word embeddings should follow the normal distribution but this does not invalidate the analysis: if they follow it, then we need to apply initialization formula (90), if they don’t — formula (6) which covers a more general case (it is derived with the relaxed assumptions).
> >
> > > H6: Histograms in Fig. 13 look quite normal.
> >
> > We respectfully disagree and they are not normal objectively speaking,  as we said in our previous response. To verify that, we used [D’Agostino test](https://en.wikipedia.org/wiki/D%27Agostino%27s_K-squared_test) which confirmed our point with a very high confidence (p-value < 0.005 for SUN/CUB/aPY and < 0.05 for AwA). This shows that with high confidence, zero-mean and unit-variance assumptions do not hold for SUN/CUB/aPY/AwA attributes
> >
> > - [1] https://arxiv.org/abs/1603.00550
> > - [2] https://arxiv.org/abs/1804.09458
> > - [3] https://arxiv.org/abs/1909.05995
> > - [4] https://arxiv.org/abs/1805.11604

---

> > ### Author Response · Authors · 2020-11-25
> > **Elaborating on some misunderstandings**
> >
> > We thank Reviewer1 for the response and we hope our detailed responses and additional experiments are showing that we are doing what is in our hands to both clarify a major misunderstanding and also to apply actionable feedback based on the right ground. If that’s alright, we hope to clarify to R1 what made us deeply concerned about his/her judgment. We of course aim to have the best version of our paper but without having our key message or contributions distorted or have much deviation from it. The reason why we think we may not on the right ground is for example the quote below that mentions a claim, we think we did not make.
> >
> > Example claims R1 attributed to our paper that we believe we did not make.
> > > The authors again claim that the main bottleneck in improving zero-shot learning is “variance control” (the end of Sect. 3.3).
> >
> > In our paper, we employed a hypothesis that “improving” the variance improves ZSL performance. This is not a new observation looking at the wider circle of supervised learning. In the context of ZSL, it is less studied and understood and that motivated our work. Following this hypothesis, we derived CN to improve the variance which in turn improved the performance. This suggests the hypothesis to be trustworthy and this is why we allowed ourselves to say that  “[AN/NS/CN] help training by preserving variance during a forward pass”.
> >
> > > In order to judge the significance of the contribution “[...]", I think one could use (i) […] (ii) [...] (iii) [...]
> >
> > First, we believe that “(i) difficulty to prove” should not influence a statement's significance. Second, we believe that apart from (ii, iii), there is a more important point to value a statement’s significance:
> >
> > - (iv) how much insight it provides to build stronger algorithms
> >
> > This motivated us to build a model that is
> > 1) very simple, and
> > 2) beats the modern SotA in terms of both scores and training speed.
> >
> > This is why we believe that (iv) very well applies to our work.
> >
> > > [...] Figures 1, 4, 5, 6, 7 and the tables shows (ii) but provides only weak empirical evidence to support (iii) (authors disagree).
> >
> > We are glad that Reviewer 1 confirmed that (ii) is shown in the paper. As to (iii), we respectfully disagree with their judgement, and our reasoning is as follows.
> > - `X leads to Y (from theory)`: Our theoretical statements demonstrate how AN/NS (i.e. `X`) influence the variance (i.e. `Y`). We validate them in practice in order to confirm that the assumptions are reasonable — see Figures 1,5,6,7,8.
> > - `X leads to Z (from extensive experiments and analysis)`: Experiments done by us and [1,2] demonstrate that AN/NS leads to better performance (see below). To confirm this even better, we ran additional experiments for 1 and 2-layer MLP and CVC-ZSL [3] (see below).
> >
> > Now, the above two points implies correlation between `Y` and `Z`: when we enable AN/NS then both the variance and the performance improves, when we disable them — both worsens. Hence, the correlation.
> >
> > Showing `Y leads to Z` *directly* is out of reach from both theoretical and empirical perspectives:
> > - Theoretical difficulties are described in one of our [previous responses](https://openreview.net/forum?id=7pgFL2Dkyyy&noteId=8MNECymkktX).
> > - Practical difficulties arise because there is no clear way of how to introduce a change in the variance without changing the architecture/parametrization/optimization process. Even the simple tricks we consider in the work influence not only the variance, but also other things as well (for example, NS makes logit scores to be computed based only on vectors directions, AN reduces the gradient scale of the first layer, CN influences the loss landscape, etc.). To the best of our knowledge, there is no existing way to isolate the variance influence.
> >
> > > Further, the tricks have been shown to work only on one zero-shot learning MLP baseline in this paper (the authors believe this is enough)
> >
> > [Existing Literature] Their importance for existing methods can be found in the related papers which we discuss in Section 2, paragraph 2. For example, [1] improved on AwA/CUB by 10/20% using AN; [2] improved their model by 10-15% using NS. Also, [3] motivated the use of NS by ``variance reduction’’ (see Section 3.1 of [3]), but didn’t provide any further exploration on that (we developed this point to a much more rigorous basis).
> >
> > [More Evidence & Ablations] To incorporate more evidence for the importance of AN and NS, we additionally ablated them for Linear/2-layer MLPs and CVC-ZSL [3] with/without AN/NS for SUN/CUB/AwA1/AwA2 datasets and reported the results in Appendix D.6 that demonstrate the improvement by ~5-50%. Also, in Table 4 there is an ablation over values of $\gamma$ for NS for Linear and 2,3 layered MLPs.
> >
> > Please, also see our additional response on F1, F2 and H1, H2, H4, H6 below.
> >
> > [1] https://arxiv.org/abs/1603.00550
> > [2] https://arxiv.org/abs/1804.09458
> > [3] https://arxiv.org/abs/1909.05995
> > [4] https://arxiv.org/abs/1805.11604

---

### Author Response · Authors · 2020-11-20
**CN paper update**

We thank the reviewers for their valuable feedback. We are encouraged that they found our method to be simple and showing strong empirical results (R1), the paper to be well organized and easy to follow (R3), empirical evidence to be convincing (R4), theoretical analysis to be interesting and sensible (R3), our proposed metrics for CZSL to be more generic and realistic (R3). We have replied in detail to address the reviewer’s concerns individually and incorporated the feedback.


Key changes we made to the paper:
- Applied our model's idea to RelationNet and CVC-ZSL methods and observed the improvement in both cases by +2.0 and +1.8 on average in terms of GZSL-H for them, respectively. We report these results in Appendix D.5 of the paper.
- Enriched the exposition on attributes normalization, normalize+scale, and embedding-based ZSL models in the Related Work section (first two paragraphs).
- Improved our paper's clarity for the smoothness and normalization exposition and added additional details for them (sections 3.3-3.4 and Appendix F).
- After submission, we found an error in our evaluation procedure for AwA1/AwA2 datasets that was leading to higher scores. We still hold the SotA result for AwA2 with GZSL-H of 67.6. For AwA1, our position descended to third place (after EPGN and CVC-ZSL) with GZSL-H of 67.8. We updated Table 1, 2 from the main body and Table 5 from Appendix D accordingly.
- Discussion of the seen class logits scaling trick in Section 5 and Appendix D.4
- Added [1, 2] to the comparison in Table 2
- Added CZSL baselines descriptions
- Added discussion of Figure 12 and Figure 13 in Appendix H
- Launched additional experiments to demonstrate the importance of AN/NS on other models: 1/2-layer MLP and CVC-ZSL (see Table 8 in Appendix D.6)
- Added exposition with the experimental validation of what happens when we manually enforce zero-mean/unit-variance attributes assumptions instead of using the formula derived with their relaxation in Appendix H
- Rewrote exposition on loss surface smoothness and moved it into Section 3.5

**Code**

Reviewer 2 asked for the source code for AwA1/AwA2 datasets and we prepared an anonymized [Colab Notebook](https://gist.github.com/iclr2021-classnorm/c29c8a1d4da78eb75a4cae24348b061d) for them. We would like to draw your attention to how simple our method is: it takes just ~150 lines of code including all the data downloading/preprocessing/training/evaluation/comments. And it takes just ~30 seconds to run!

Reviewer 3 and Reviewer 4 also questioned how our method would perform on aPY and thus we launched the same [Colab Notebook](https://gist.github.com/iclr2021-classnorm/cebb4dd7ca9ccc9a91d18b84aee056f7) (without changing a single hyperparameter) on this dataset. We obtained GZSL-H of 38.9 which puts our method in third place after DVBE (GZSL-H = 41.8) and CVC-ZSL (GZSL-H = 39.0), performing is very similar to CVC-ZSL. It took just 8 seconds to obtain that high score! We believe that after hyperparameter tuning the result could be improved.

- [1] Learning to Compare: Relation Network for Few-Shot Learning, CVPR=18
- [2] Meta-Learning for Generalized Zero-Shot Learning

We demonstrated in our paper a rigorously validated exploration of how our proposed CN can significantly improve simple ZSL  models to be on par and sometimes surpassing the state-of-the-art. In Continual / Multitask learning setting, we showed that CN can significantly improve the performance on sequential, A-GEM, MAS, and multi-task methods (Table 3).

---

### Decision · Program_Chairs · 2021-01-07
**Final Decision**

**Decision:**

Accept (Poster)

**Comment:**

This paper got mixed reviews. One for reject and three for acceptance. The reviewers and authors have extensive discussion. Authors also provided additional experiments for further clarifying some questions from the reviewers. The paper has some clarify issue in the theoretical justification part as pointed out by AR1. Authors should extensively improve this part or revise the statement. However, the method proposed in this paper is simple and the results are indeed good. This paper is valuable and should be shared within the community to advance research on ZSL. Therefore, AC recommends acceptance.